# SpikeGen: Decoupled "Rods and Cones" Visual Representations Processing with Latent Generative Framework

**Gaole Dai**[*]  **Menghang Dong**[*]  **Rongyu Zhang**[*]  **Ruichuan An**

**Tiejun Huang**[✉]
tjhuang@pku.edu.cn

**Shanghang Zhang**[✉]
shanghang@pku.edu.cn

State Key Laboratory of Multimedia Information Processing, School of Computer Science,
Peking University

## Abstract

The process through which humans perceive and learn visual representations in dynamic environments is highly complex. From a structural perspective, the human eye decouples the functions of cone and rod cells: cones are primarily responsible for color perception, while rods are specialized in detecting motion, particularly variations in light intensity. These two distinct modalities of visual information are integrated and processed within the visual cortex, thereby enhancing the robustness of the human visual system. Inspired by this biological mechanism, modern hardware systems have evolved to include not only color-sensitive RGB cameras but also motion-sensitive Dynamic Visual Systems, such as spike cameras. Building upon these advancements, this study seeks to emulate the human visual system by integrating decomposed multi-modal visual inputs with modern latent-space generative frameworks. We named it *SpikeGen*. We evaluate its performance across various spike-RGB tasks, including conditional image and video deblurring, dense frame reconstruction from spike streams, and high-speed scene novel-view synthesis. Supported by extensive experiments, we demonstrate that leveraging the latent space manipulation capabilities of generative models enables an effective synergistic enhancement of different visual modalities, addressing spatial sparsity in spike inputs and temporal sparsity in RGB inputs.

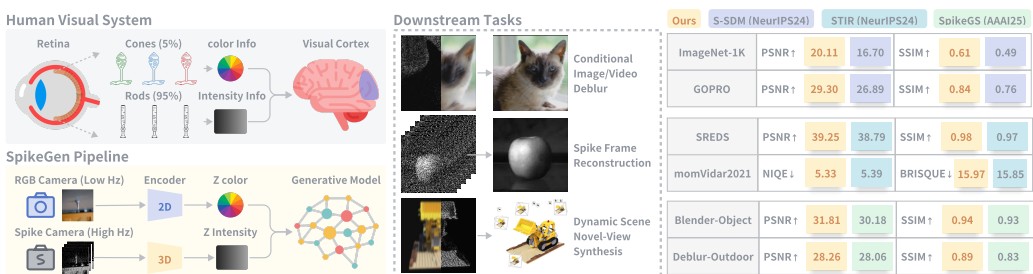

Figure 1: Overview of the motivation, task setups and main quantitative results of SpikeGen.

---

[*]Equal Contribution, [✉]Corresponding Author

# 1 INTRODUCTION

Human vision is not the best at any single metric — bees can be tetrachromats (Kevan et al., 2001) and many flies see flicker at far higher rates than we do (Borst et al., 2010) — but it excels as a balanced, tightly coupled system that fuses rich low-level cues from different modalities with high-level semantics. This balance yields **(i)** broad color discrimination via trichromacy with fine color resolution (Horiguchi et al., 2013), **(ii)** robust perception of moving targets with preserved spatial detail through active eye movements and temporally tuned pathways (Conner, 1982), and **(iii)** powerful inference mechanism that "fills in" missing information under extreme conditions (e.g., low light, fast motion) (Chariker et al., 2016; 2018).

The quest to replicate such capabilities has driven innovations in both imaging hardware and computational models. In hardware, RGB cameras directly emulate the trichromatic properties of cone cells (Peterson, 2016), which are concentrated in the fovea to provide high spatial and color resolution but lack temporal sensitivity (Zhang et al., 2021a). Conversely, Dynamic Visual Systems (DVS) (Han et al., 2020) like spike cameras draw inspiration from rod cells — widely distributed outside the fovea — utilizing their continuous integration mechanism to achieve high temporal resolution for perceiving subtle light changes, albeit with trade-offs in color and spatial resolution (Marković et al., 2020). On the algorithmic front, Artificial Neural Networks (ANNs) have emerged as powerful tools for multimodal visual fusion. For instance, infrared imaging can detect heat sources in low-light but lacks color and texture details, while visible-light imaging provides high-resolution color information but falters in darkness. Fusion algorithms integrating these modalities have demonstrated significant utility in applications like nighttime security (Yuan et al., 2024).

Notably, the RGB and spike modalities are functionally decoupled. In the human visual system, the brain acts as a "model" that integrates information from both modalities while leveraging semantic knowledge to infer missing details. Inspired by this mechanism, our approach aims to develop a multimodal model that emulates key characteristics of the human visual system. Reviewing current studies, when relying solely on the RGB modality, although achieving relative improvements over low - quality inputs, often results in inherent information loss that introduces ambiguities in the output (Chen et al., 2024; Chang et al., 2023; Chen et al., 2023a). This phenomenon is known as the "sharpness trap" where overall contrast is enhanced without meaningful geometric recovery (i.e., preservation of fine structural features) (Jiang et al., 2021). Similarly, in tasks dominated by the spike modality — such as dense frame reconstruction — the outputs are typically grayscale and exhibit limited spatial resolution (e.g., 250×400). As a result, incorporating priors from complementary modalities can significantly improve performance. Prior work, such as the Self-supervised Spike-guided Deblurring Model (S-SDM) (Chen et al., 2024), has demonstrated that texture cues captured by spike cameras can guide models to prioritize structural accuracy over superficial sharpness. Furthermore, we argue that even blurry RGB frames may provide valuable guidance for dense frame reconstruction. Since individual frames in the raw spike stream are spatially sparse, insufficient spike activity leads to spatial uncertainty (Zhu et al., 2019; 2020; Zheng et al., 2023). In contrast, RGB frames preserve global spatial relationships among scene elements, thereby serving as a coarse yet effective constraint to mitigate such uncertainty. In conclusion, cross-modality processing exhibits greater potential, with recent successes extending to 3D vision tasks (e.g., multi-view spike stream for 3DGS (Guo et al., 2025; Dai et al., 2024)) further validating this trend.

Nevertheless, we observed a significant discrepancy between current approaches — such as S-SDM — and the human visual system. While existing methods rely on pixel-level self-supervised learning (SSL), the human visual system acquires information through latent representations without explicit pixel reconstruction. To bridge this gap, we propose *SpikeGen*, a model featuring a latent generative architecture, for the following reasons: **(i)** Since the emergence of Latent Diffusion Models (LDM) (Rombach et al., 2022), operations in latent space have become an efficient paradigm for model training. In SpikeGen, we pre-train the spike encoder to align with the latent space of RGB Variational Autoencoder (VAE) (Kingma et al., 2013), achieving an 512x spatial-temporal downsampling. This substantially reduces computational overhead during training, particularly in terms of pixel-space loss computation, thereby enabling efficient pre-training on large-scale synthetic datasets (Deng et al., 2009). **(ii)** Pre-training in the latent space also improves adaptability to downstream tasks. Although pixel-level SSL methods such as Masked Autoencoders (MAE) (He et al., 2022) have proven effective, recent advances in DINO (Caron et al., 2021; Oquab et al., 2023) and Joint-Embedding Prediction Architectures (JEPA) (Assran et al., 2023; Drozdov et al., 2024) demonstrate superior generalization

by operating in the embedding (i.e., latent) space. A key commonality among these approaches is their focus on capturing latent-level similarities rather than minimizing pixel-wise reconstruction errors, which helps mitigate issues such as the "sharpness trap." **(iii)** Most prior works are based on deterministic modeling frameworks (Chen et al., 2024; 2023a; Chang et al., 2023; Fan et al., 2024). In contrast, SpikeGen adopts a probabilistic framework by performing diffusion in the VAE latent space. This choice is motivated by that the scenarios we address inevitably involve information loss, whether due to blurring in RGB modalities or spatial sparsity in spike modalities. Diffusion models exhibit superior performance in similar cases, such as super-resolution (Moser et al., 2024; Gao et al., 2023) and denoising (Ho et al., 2020; Kawar et al., 2022), attributed to their enhanced recovery accuracy, particularly for detailed content like textures and artificial structures.

In summary, our key contributions are as follows:

1. We systematically reviewed the merits of the human visual system, identifying the functionally decoupled nature of the human eye and cortical processing in latent space as fundamental to this advancement. Inspired by these biological mechanisms, we propose a novel processing pipeline compatible with dual RGB-spike modality, named ***SpikeGen***.

2. To the best of our knowledge, our work represents a pioneering effort in leveraging a latent-based generative model for such tasks. The rationale stems from both **efficiency and effectiveness** considerations: latent-space operations enable an exceptional compression ratio (512-fold) during training while circumventing common issues with pixel-space loss, such as the "sharpness trap," which undermines the model's generalization capability.

3. We validate SpikeGen across multiple downstream tasks, including conditional image/video deblurring, dense frame reconstruction from spike streams, and high-speed scene novel-view synthesis. This comprehensive evaluation encompasses **all major tasks** in visual spike stream & RGB processing, thereby strongly supporting our hypothesis regarding the challenges inherent to these tasks and the corresponding methodological designs.

## 2 RELATED WORKS

**Visual Spike Stream Processing**     Spike cameras are bio-inspired vision sensors that capture local intensity changes asynchronously, emitting spike streams. Compared to traditional frame-based cameras, they offer advantages such as high dynamic range (HDR), low power consumption, high temporal resolution, and inherent data compression, making them ideal for high-speed dynamics and challenging lighting conditions. Early research focused on texture recovery methods, like retina-inspired sampling by Zhu et al (Zhu et al., 2019). More recently, deep learning has dominated the transformation of spike data into dense images. Zhao et al. introduced Spk2ImgNet (Zhao et al., 2021) for reconstructing dynamic scenes from spike streams. Spiking Neural Networks (SNNs) (Zhang et al., 2023b) have also been applied for biologically plausible processing, with Zhao et al. developing SSIR (Zhao et al., 2023) architectures. Fan et al. proposed STIR (Fan et al., 2024) for spatio-temporal interactive learning to improve reconstruction efficiency. In image deblurring, Chen et al. developed SpkDeblurNet (Chen et al., 2023b)and S-SDM (Chen et al., 2024) to enhance motion deblurring using spike information. Spike cameras are also advancing 3D vision and scene perception, as shown by Dai et al (Dai et al., 2024) (SpikeNVS) and Guo et al. (Guo et al., 2025)(SpikeGS).

**Latent Generation Models**     Generative modelling has made significant progress, enabling high-fidelity image synthesis and other complex data modalities. Diffusion models, pioneered by Denoising Diffusion Probabilistic Models (DDPMs) Ho et al. (2020), achieve state-of-the-art results by progressively denoising signals from Gaussian noise. To address slow sampling speeds, Song et al. Song et al. (2020) introduced Denoising Diffusion Implicit Models (DDIMs), which offer faster generation with comparable quality. Latent Diffusion Models (LDMs) Rombach et al. (2022) operate in a learned latent space, reducing computational demands. SDXL Podell et al. (2023) further improved LDMs for high-resolution image generation. Auto-regressive models also benefit from latent representations; MaskGIT Chang et al. (2022) uses transformers for masked generative image modelling with discrete latent codes. Recent research Li et al. (2024) explores autoregressive generation directly from continuous features, eliminating the need for quantization.

**Self-Supervised Learning**  Self-Supervised Learning (SSL) has long been a prominent research topic. The success of MAE (He et al., 2022) and SimMIM (Xie et al., 2022) demonstrated the effectiveness of Masked Image Modeling (MIM) in pixel space, prompting recent SSL approaches to investigate MIM in latent space as a more efficient and effective alternative. Image Joint-Embedding Prediction Architectures (i-JEPA) (Assran et al., 2023) represent a representative effort in this direction. Concurrently, DINO (Caron et al., 2021) explored the potential of contrastive learning (CL) in latent space by aligning semantically consistent representations of the same image under different augmentations. iBot (Zhou et al., 2021) integrated the strengths of latent space modeling, MIM, and CL into a unified framework, inspiring subsequent works such as DINOv2 (Oquab et al., 2023) and DINOv3 (Siméoni et al., 2025), which have scaled up latent space SSL to the billion-parameter level in terms of both data volume and model capacity.

## 3 METHODS

### 3.1 PRELIMINARY

**Visual Spike Stream**  The imaging principle of spike cameras diverges from both the exposure-based mechanism of conventional RGB cameras and the differential approach of event cameras. A spike camera operates by integrating incoming light intensity until an accumulator reaches a predefined activation threshold, denoted as $V_{th}$. Upon reaching this threshold, a spike is emitted, and any surplus intensity $I$ beyond the threshold is retained for the next integration cycle. If $I_t$ represents the light intensity input at time step $t$, the value stored in the accumulator, $A_t$ evolves according to:

$$A_t = (A_{t-1} + I_t) \bmod V_{th} \tag{1}$$

A spike for a pixel $p$ at coordinates $(i, j)$ occurs when the accumulated value plus the current input signal meets or exceeds the threshold $V_{th}$. This determines the binary spike value, indicating whether the brightness level was sufficient during the sampling interval. The formal definition is:

$$p_{i,j,t} = \begin{cases} 1, & \text{if } A_{t-1} + I_t \geq V_{th} \\ 0, & \text{otherwise} \end{cases} \tag{2}$$

Reconstructing dense frames from these spike streams traditionally employs methods like Texture From Interval (TFI) or Texture From Playback (TFP) (Zhu et al., 2019). TFI excels at capturing texture contours. In contrast, TFP reconstructs textures across varied dynamic ranges by adaptively modifying the time window size based on contrast levels. Let $d_{i,j,t}$ be the temporal latency, i.e., the time elapsed since the last spike at pixel $(i, j)$ before time $t$. Let $w$ denote the size of the time window and $N_{i,j,w}$ be the total number of spikes accumulated for pixel $(i, j)$ within that window. The expressions for TFI and TFP reconstructed texture $P^t$ at time $t$ are:

$$\textbf{TFI: } P^t_{TFI} = \frac{V_{th}}{d_{i,j,t}}, \quad \textbf{TFP: } P^t_{TFP} = \frac{N_{i,j,w}}{w} * C \tag{3}$$

where $C$ is a constant scaling factor for TFP.

**Latent Space Operation**  Latent Diffusion Models (LDMs) (Rombach et al., 2022) are a type of latent space generative model designed for high-resolution image synthesis while being computationally efficient. Instead of operating directly in the high-dimensional pixel space like traditional Diffusion Models, LDMs apply the diffusion process within the lower-dimensional latent space learned by a powerful pre-trained autoencoder (e.g., VAE (Kingma et al., 2013)). This autoencoder consists of an encoder $\mathcal{E}(\cdot)$ that compresses the image $x$ into a latent representation $z = \mathcal{E}(x)$ and a decoder $\mathcal{D}(\cdot)$ that reconstructs the image from the latent space $\tilde{x} = \mathcal{D}(z)$ (Rombach et al., 2022). The diffusion model $\epsilon_\theta$ is then trained solely within this latent space to denoise representations $z_t$ at various noise levels $t$. This separation allows the computationally expensive training and sampling of the diffusion process to occur in a much more manageable space, focusing on semantic information rather than imperceptible pixel details.

The training objective for the Latent Diffusion Model is given by:

$$L_{LDM} := \mathbb{E}_{\mathcal{E}(x), \epsilon \sim \mathcal{N}(0,1), t}[||\epsilon - \epsilon_\theta(z_t, t)||_2^2]$$

Here, $\epsilon$ is the noise sampled from a normal distribution, $z_t$ is the noisy latent representation at timestep $t$, and $\epsilon_\theta(z_t, t)$ is the neural network predicting the noise added to $z_t$.

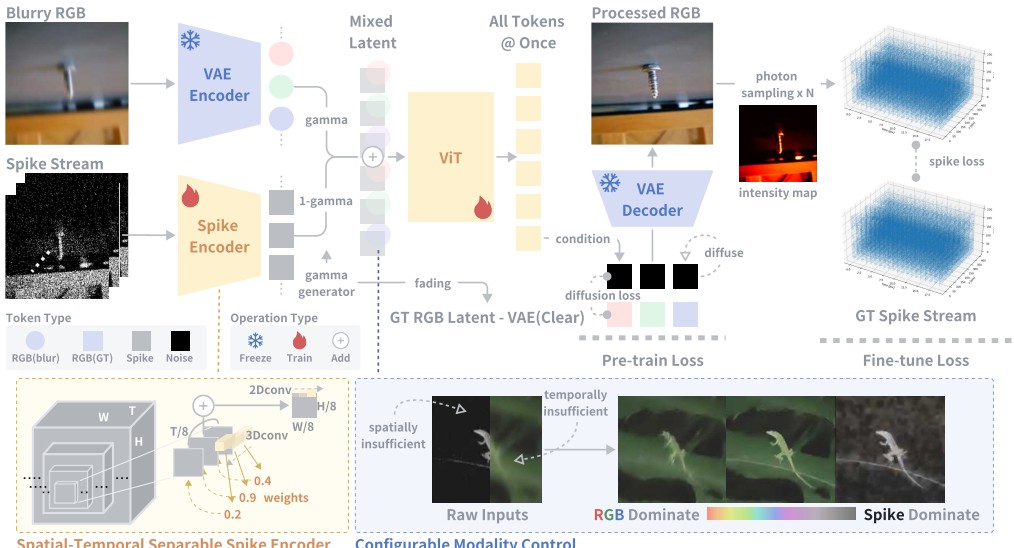

Figure 2: **The Overall Pipeline of SpikeGen.** SpikeGen adopts a standard pre-training (self-supervised) and fine-tuning (task-dependent) pipeline. Specifically, visual information from two modalities is encoded, followed by the addition of the two latent representations. In this process, $\gamma$ serves as a parameter to control the effective weight and is randomly sampled from the interval [0, 1]. Notice that during the pre-training phase, the diffusion loss is computed using the pre-extracted latent representation of the clear RGB image obtained via a Variational Autoencoder (VAE) (Kingma et al., 2013). The spike stream loss during the fine-tuning phase is calculated based on the given ground truth and the synthetic spike stream generated from the predicted RGB output.

## 3.2 SPIKEGEN: CONFIGURABLE DUAL MODALITY PRE-TRAIN

SpikeGen is developed by leveraging recent advancements in latent generation models. Specifically, we constructed a modified version of the Masked Auto-Regressive Model (MAR) (Li et al., 2024) to better suit our tasks. Similar to MAR, we encode RGB visual information via a Variational Autoencoder (VAE) (Kingma et al., 2013) and generate the conditioning input using a standard Vision Transformer (ViT) (Dosovitskiy et al., 2020) backbone. Subsequently, the conditioning input is processed through a compact Multi-Layer Perceptron (MLP), conducting a per-token diffusion process (Song et al., 2020) for decoding the latent representation (see Figure 2).

**Diffusion with Decomposed Latent Condition** From a broader view, unlike the Masked Image Modelling (MIM) (He et al., 2022) strategies employed in MAR, we consider that both blurry RGB inputs (temporally insufficient) and spike inputs (spatially insufficient) represent forms of degradation. Consequently, the ViT in SpikeGen receives complete tokens from two encoders and generates conditions for the per-token diffusion. Since we do not need to predict new tokens from void, we have further streamlined the auto-regressive process of MAR into generating all tokens simultaneously. This enhances the efficiency during both training and inference (see Appendix Figure 6). We also found that this design did not affect the overall performance (see Appendix Table 7)

**Spatial-Temporal Separable Spike latent** Besides the Variational Autoencoder (VAE) utilized for encoding blurry RGB information, we developed a Spatial-Temporal Separable Spike (S3) Encoder. The S3 Encoder first applies a series of 3D convolutional blocks to transform the input spike stream from dimensions $[B, 1, T, H, W]$ to $[B, C_{out}, T/8, H/8, W/8]$ ($C_{out} = 512$). These blocks progressively reduce the spatio-temporal resolution while increasing the channel depth, akin to the hierarchical structure employed in the UNet encoder. Following the temporal fusion stage, where the $[B, C_{out}, T/8, H/8, W/8]$ features are processed to fuse information along the temporal dimension explicitly, we generate temporal attention weights for the features using two consecutive 1x1x1 3D convolutions. These attention weights are then multiplied element-wise with the features.

The resulting weighted features are subsequently summed along the temporal dimension, collapsing it and producing a feature map of shape $[B, C_{out}, H/8, W/8]$. To further refine these spatially-resolved but temporally-fused features, a 2D convolution is applied, followed by a LayerNorm operation and a final LeakyReLU activation. When temporal fusion is active, the ultimate output of the S3 Encoder is a feature map of shape $[B, C_{out}, H/8, W/8]$ (see Figure 2 and Appendix Table 5).

**Random Modality Dropout** To enable a configurable modality control after pre-training, we randomly assign the addition ratio of RGB latent (extracted by VAE encoder) and the spike latent (extracted by S3 encoder). The random ratio is denoted as $\gamma$, which is sampled from a Gaussian distribution with a mean of $\mu = 0.5$ and a variance of $\sigma^2 = 1$, subsequently truncated to the interval $[0, 1]$. This can be formally expressed as $\gamma \sim \mathcal{N}_{[0,1]}(\mu = 0.5, \sigma^2 = 1)$. The mixed latent, $z_{mixed}$, can then be calculated by the formula $z_{mixed} = (1 - \gamma)z_{RGB} + \gamma z_{spike}$. We intend to simplify the mixing strategy to enable better downstream usage. However, this indicates we cannot directly use the latent extracted from the clear RGB images as learning objectives. Instead, we colour fade the clear RGB images based on the $\gamma$ value (higher fade intensity with increasing $\gamma$). Let $I_{clear}$ represent the original clear RGB image and $I_{gray}$ be its corresponding grayscale version. The fade formula to obtain the faded image $I_{faded}$ is as $I_{faded} = (1 - \gamma) \cdot I_{clear} + \gamma \cdot I_{gray}$ This ensures that when $\gamma \to 1$ (spike latent becomes dominant), $I_{faded}$ tends towards $I_{gray}$, and when $\gamma \to 0$ (RGB latent is dominant), $I_{faded}$ remains close to $I_{clear}$. This enables, when spike latent becomes the main content ($\gamma$ is high), the model to focus more on texture reconstruction over precise colour prediction, as the learning objective $I_{faded}$ will have reduced colour information (see Figure 2).

## 3.3 SpikeGen: Task Adaptation

During pre-training, our loss is computed solely as the diffusion loss between the clear RGB latent and the latent predicted by SpikeGen. This significantly reduces computational costs when training on large-scale datasets. However, this latent alignment can underperform during fine-tuning, particularly with limited data, as it may not sufficiently guide the model to capture fine-grained details. For instance, the outdoor dataset (Ma et al., 2022) for 3D scene reconstruction contains merely 34 images per scene. A common strategy to mitigate this is to introduce pixel-space similarity measures during the fine-tuning phase; for example, models like SDXL (Podell et al., 2023) incorporate both MSE and perceptual losses calculated in the RGB pixel space. However, as downstream tasks for SpikeGen could lack clear RGB ground truth, we propose a spike-alignment strategy instead (see Figure 2). Specifically, the latent generated by SpikeGen is decoded back into the pixel space, yielding a predicted image $I_{pred}$. Based on the intensity of $I_{pred}$, we then generate a corresponding probability map $P_{pred}$. This process starts with min-max normalization and the $I_{norm}$ image is convolved with a Gaussian kernel $K_G$ (parameterized by $\sigma_s$) to produce a smoothed version $I_{smooth}$. Finally, $I_{smooth}$ undergoes gamma correction, $P_{pred} = (I_{smooth})^{\gamma_c}$, with a small amount of uniform random noise is added. Predicted spike stream can then be generated by sampling $P_{pred}$. The spike-alignment loss is computed by comparing the ground truth spike stream with the predicted spike stream.

## 4 Experiments

We conducted comprehensive comparisons with more than 20 state-of-the-art baselines across 3 major visual spike & RGB processing tasks (conditional image/video deblurring, dense frame reconstruction from spike streams, and high-speed scene novel-view synthesis) to demonstrate the effectiveness and versatility of SpikeGen. We color-coded the performance in Table 1, 2, 3 with **Red (1st)**, Blue (2nd). All the experiments follow the data usage and evaluation strategies used in S-SDM (Chen et al., 2024), STIR (Fan et al., 2024), and SpikeGS (Guo et al., 2025). See Appendix A for more details.

## 4.1 Pre-training

We performed pre-training on the complete training set of ImageNet (Deng et al., 2009) utilizing 8 A800 GPUs. We carefully tuned the hyperparameters to optimize the generation of synthetic data. Given the substantial volume of available data, we adopted a more aggressive blurring configuration and a sparser spike stream to enhance the learning challenge for the model. Specifically, we applied a $40 \times 40$ blurring kernel and random sampled 8 from 64 generated spike frames per image. Through extensive pre-training, we anticipate that SpikeGen will exhibit robust generalization capabilities and

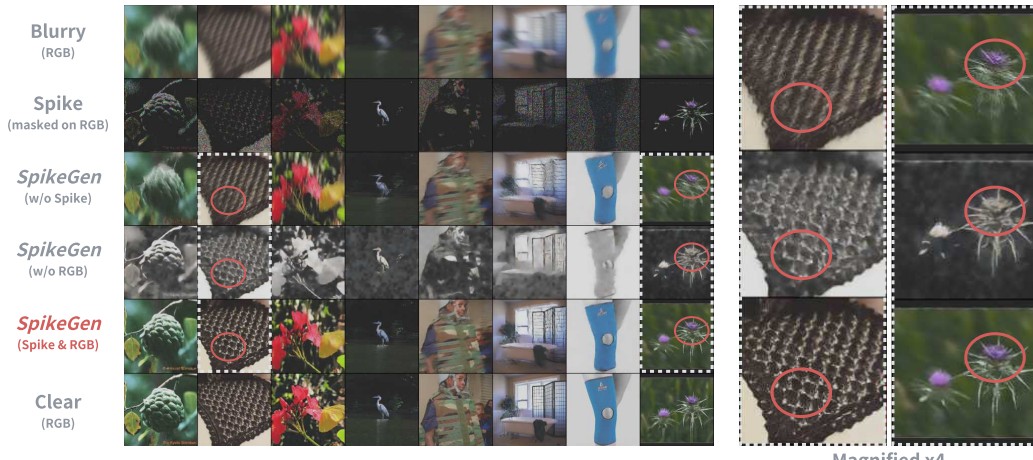

Figure 3: **Conditional Image Deblurring on Synthetic RGB-Spike Data.** For all our experiments, the input visual spike streams for SpikeGen are in binary format (0/1) without colour information. Here, for better visualization, row 2 (from top) of the left panel demonstrates the cut-out result of the RGB channels using 3 spike frames (results in row 4 used 8 frames as input). We also magnified a few results with highlighted detail for better comparison of the structural correctness (right panel).

successfully achieve the configurable modality conditioning discussed in Section 3.2. In Figure 3, we illustrate the generalization performance of SpikeGen on the ImageNet test set. Overall, even when RGB images are significantly blurred and spike information is relatively sparse, SpikeGen can effectively reconstruct artificial geometric shapes of objects (column 1, right) and natural fine structures (column 2, right). Furthermore, by controlling the injection of modalities, we showcased single-modal generalization capability (rows 3 and 4, left). When dual modalities collaborate, the model leverages their complementary strengths to achieve optimal results (row 5, left). We provided more visualization in Appendix C.2.

Furthermore, in terms of efficiency, we aim to evaluate whether the latent generation backbone selection in SpikeGen, combined with a non-autoregressive design (see Section 3.2), can reduce computational overhead. As shown in Table 7, our approach achieves a significant reduction in processing time without compromising performance.

### 4.2 FINE-TUNING GENERALIZATION

Given the extensive range of downstream experiments we have conducted, this section presents only the most concise and comparative results. For additional details, please refer to Appendix C.

**Conditional Image/Video Deblurring**  We adhered to the same experimental setups as described in S-SDM (Chen et al., 2024) for this task. S-SDM generated paired blurry RGB-spike stream data based on the GOPRO dataset (Nah et al., 2017). Each blurry RGB input was converted by the SpikingSim simulator (Zhao et al., 2022) into an average of 98 spike frames (we use 8) with a spike threshold $V_{th} = 1$ during training and $V_{th} = 1/2/4$ during validation. As shown in the quantitative results presented in Table 1, SpikeGen significantly outperformed all baselines across all $V_{th}$ configurations. Specifically, the relative improvement in PSNR increased from approximately 1 to approximately 3 as the spike guidance became sparser (higher $V_{th}$). We attribute this phenomenon to two primary factors. First, the pre-training to fine-tuning strategy we adopted enabled us to leverage the diversity of data encountered during self-supervised learning in latent space. Second, the random modality dropout applied during pre-training enhanced robustness when dealing with limited spike guidance. Additionally, the use of fewer spike frames at this stage further strengthened this capability (see Appendix C.3 for ablation studies using pure RGB inputs). This observation is also corroborated by the qualitative results depicted in Figure 4.

Table 1: **Quantitative Results of Conditional Video Deblur Task.** The column colour reflects different thresholds when generating spike frames (code and value from S-SDM (Chen et al., 2024)). Higher thresholds produce spike frames with higher sparsity.

| Methods | Dual Modality | $V_{th} = 1$ | | $V_{th} = 2$ | | $V_{th} = 4$ | |
|---|---|---|---|---|---|---|---|
| | | PSNR ↑ | SSIM ↑ | PSNR ↑ | SSIM ↑ | PSNR ↑ | SSIM ↑ |
| *LEVS* (CVPR18) (Jin et al., 2018) | ✗ | 21.16 | 0.60 | 21.16 | 0.60 | 21.16 | 0.60 |
| *Motion-ETR* (TPAMI21) (Zhang et al., 2021b) | ✗ | 21.96 | 0.61 | 21.96 | 0.61 | 21.96 | 0.61 |
| *BiT* (CVPR23) (Zhong et al., 2023) | ✗ | 23.64 | 0.70 | 23.64 | 0.70 | 23.64 | 0.70 |
| *TRMD* (TMM24) (Chen & Yu, 2024) | ✓ | 27.32 | 0.78 | 21.20 | 0.60 | 18.57 | 0.52 |
| *RED* (ICCV21) (Xu et al., 2021) | ✓ | 24.46 | 0.74 | 23.18 | 0.67 | 21.94 | 0.61 |
| *REFID* (CVPR23) (Sun et al., 2023) | ✓ | 28.12 | 0.82 | 15.29 | 0.34 | 13.62 | 0.27 |
| *SpkDeblurNet* (NIPS23) (Chen et al., 2023b) | ✓ | 28.31 | 0.83 | 14.41 | 0.30 | 11.62 | 0.20 |
| *S-SDM* (NIPS24) (Chen et al., 2024) | ✓ | 26.89 | 0.76 | 26.37 | 0.74 | 25.43 | 0.70 |
| **SpikeGen** (RGB&Spike) | ✓ | **29.30** | **0.84** | **28.78** | **0.82** | **28.07** | **0.81** |

Table 2: **Quantitative Results of Dense Frame Reconstruction Task.** The row color indicates the method type, with red denoting training-free methods, yellow representing event-based methods, and blue corresponding to spike-based methods.

| Methods | Dataset: *SREDS* (Zhao et al., 2023) | | | | | Dataset: *momVidar2021* (Zhu et al., 2020) | |
|---|---|---|---|---|---|---|---|
| | PSNR ↑ | SSIM ↑ | LPIPS ↓ | NIQE ↓ | BRISQUE ↓ | NIQE ↓ | BRISQUE ↓ |
| *TFP* (ICME19) (Zhu et al., 2019) | 25.35 | 0.69 | 0.26 | 5.97 | 43.07 | 9.34 | 45.20 |
| *TFI* (ICME19) (Zhu et al., 2019) | 18.50 | 0.64 | 0.26 | 4.52 | 44.93 | 10.10 | 58.31 |
| *TFSTP* (CVPR21) (Zheng et al., 2021) | 20.68 | 0.62 | 0.28 | 5.35 | 51.70 | 10.92 | 64.57 |
| *ET-Net* (ICCV21) (Weng et al., 2021) | 34.57 | 0.94 | 0.05 | 3.40 | 17.16 | 6.51 | 17.39 |
| *HyperE2VID* (TIP24) (Ercan et al., 2024) | 36.37 | 0.95 | 0.05 | 3.13 | 16.77 | 6.306 | 17.02 |
| *SSIR* (TCSVT23) (Zhao et al., 2023) | 32.61 | 0.92 | 0.05 | 3.47 | 15.66 | 5.75 | 25.34 |
| *Spk2ImgNet* (CVPR21) (Zhao et al., 2021) | 36.13 | 0.95 | 0.03 | 3.08 | 15.35 | 5.66 | 16.52 |
| *WGSE* (AAAI23) (Zhang et al., 2023a) | 37.44 | 0.96 | 0.02 | 3.03 | 15.56 | 5.62 | 16.15 |
| *STIR* (NIPS24) (Fan et al., 2024) | 38.79 | 0.97 | 0.02 | 2.92 | 14.84 | 5.39 | **15.85** |
| **SpikeGen** (TFP&Spike) | **39.25** | **0.98** | **0.01** | **2.83** | 14.99 | **5.33** | 15.97 |

Table 3: **Quantitative Results of Novel View Synthesis Task.** The column colour varies according to the types of datasets.

| Methods | Dual Modality | Objects (Mildenhall et al., 2021) | | | Out-Door (Ma et al., 2022) | | | Average | | |
|---|---|---|---|---|---|---|---|---|---|---|
| | | PSNR ↑ | SSIM ↑ | LPIPS ↓ | PSNR ↑ | SSIM ↑ | LPIPS ↓ | PSNR ↑ | SSIM ↑ | LPIPS ↓ |
| *3DGS* (Clear) (Kerbl et al., 2023) | ✗ | 33.31 | 0.96 | 0.05 | 30.27 | 0.91 | 0.10 | 31.79 | 0.94 | 0.07 |
| *3DGS* (Blur) (Kerbl et al., 2023) | ✗ | 26.95 | 0.88 | 0.12 | 23.38 | 0.69 | 0.45 | 25.16 | 0.78 | 0.28 |
| *DeblurNeRF* (CVPR22) (Ma et al., 2022) | ✗ | 23.71 | 0.84 | 0.19 | **28.77** | 0.86 | 0.14 | 26.24 | 0.85 | 0.17 |
| *DeblurGS* (PACMCGIT24) (Chen & Liu, 2024) | ✗ | 28.47 | 0.91 | 0.09 | 23.55 | 0.70 | 0.41 | 26.01 | 0.81 | 0.25 |
| *SpikeNeRF* (ICME24) (Guo et al., 2024) | ✓ | 28.24 | 0.92 | 0.08 | 18.14 | 0.62 | 0.41 | 23.19 | 0.77 | 0.25 |
| *SpikeGS* (AAAI25) (Guo et al., 2025) | ✓ | 30.18 | 0.93 | 0.08 | 28.06 | 0.83 | 0.17 | 29.12 | 0.88 | 0.13 |
| **SpikeGen** (3DGS) | ✓ | **31.81** | **0.94** | **0.07** | 28.26 | **0.89** | **0.13** | **30.04** | **0.92** | **0.10** |

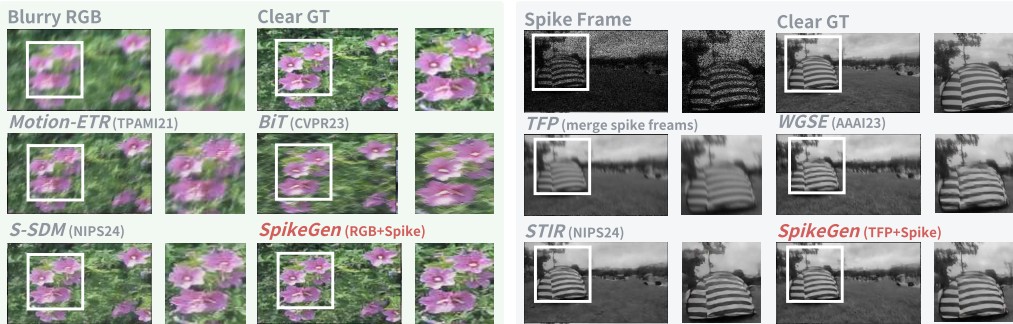

Figure 4: **Mutual Guidance with Dual Modality Inputs.** The top panel presents the results obtained from both RGB-based deblurring methods and spike-RGB-based approaches. SpikeGen demonstrated superior performance compared to all competitors in terms of visual fidelity. The bottom panel illustrates the outcomes of various methods when only a limited number of spike frames (here, 16 frames) are available. SpikeGen addresses spatial ambiguity caused by spike sparsity by leveraging the merged result of spike frames (i.e., TFP) as a pseudo-dense modality.

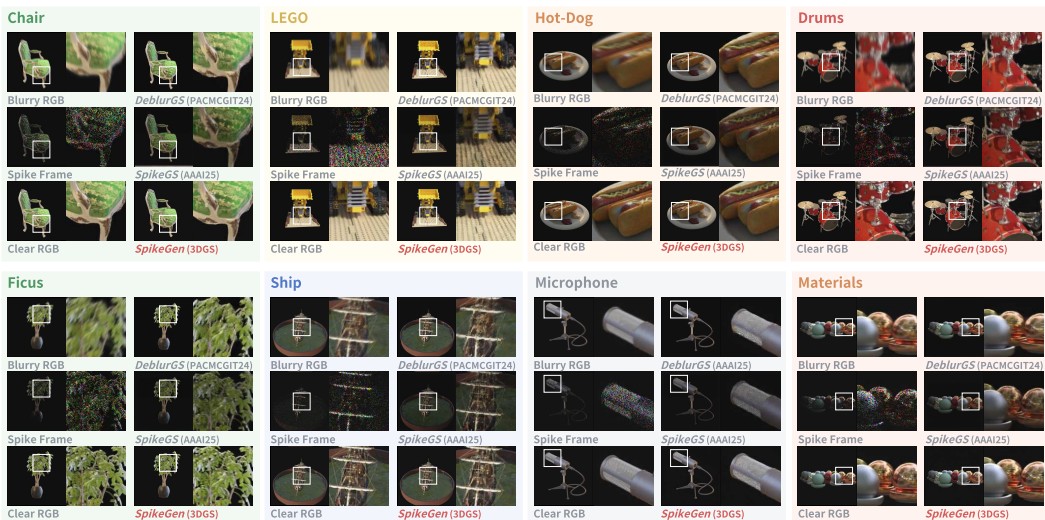

Figure 5: **Qualitative Results of Novel View Synthesis Task.**

**Dense Frame Reconstruction** As elaborated in Section 1, we posit that the spatial relationships embedded within dense RGB data, even when blurred, can address the sparsity inherent in spike streams. Consequently, validating this hypothesis on the dense frame reconstruction task holds merit. Following the experimental setups outlined in STIR (Fan et al., 2024), we utilized data from REDS (Xu et al., 2021) to synthesize a spike-based dataset (SREDS). Specifically, SREDS converts each RGB frame in REDS into 64 corresponding spike frames (we use 16) and further crops them into patches of size $96 \times 96$, resulting in a total of 21,840 patches. Additionally, the model was evaluated on the real-world dataset momVidarReal2021 (Zhu et al., 2020). As demonstrated in Table 2 and Figure 4, SpikeGen exhibits superior performance, both in referenced metrics (PSNR, SSIM, and LPIPS (Zhang et al., 2018)) and no-reference metrics (NIQE (Mittal et al., 2012b) and BRISQUE (Mittal et al., 2012a)). A pivotal factor contributing to these results is the integration of Texture from Playback (TFP) (Zhu et al., 2019) as a pseudo grayscale image. TFP aggregates all spike frames within a fixed temporal window, akin to simulating shutter timing in conventional RGB cameras for spike streams. This approach enhances spatial richness at the expense of temporal resolution, thereby introducing blur. SpikeGen mitigates this limitation by reprocessing the raw spike

streams to achieve high-quality reconstructions (see Appendix C.4 for ablation studies without TFP priors).

**High-speed Scene Novel-View Synthesis**    Finally, we evaluated SpikeGen's performance on the recently emerging task of high-speed scene novel-view synthesis. Our experimental setups were based on SpikeGS (Guo et al., 2025), which transformed the synthetic Blender datasets introduced in NeRF (Mildenhall et al., 2021) (for object-centric scenes) and DeblurNeRF (Ma et al., 2022) (for outdoor scenarios) into dual-modality pairs. As illustrated in Figure 5, while DeblurGS (Chen & Liu, 2024) demonstrates overall sharpness, it exhibits blurry details due to its reliance solely on the RGB modality. SpikeGS successfully recovers finer details; however, its optimization prioritizes texture repair using the binary spike stream at the expense of accurate color saturation matching to the ground truth RGB. In contrast, SpikeGen achieves an excellent qualitative balance and further validates its adaptability through quantitative results (see Table 3). Notably, despite being a two-stage method, SpikeGen outperforms other approaches, as evidenced by extra experiments comparing various two-stage methods (see Appendix C.5).

### 4.3   CONCLUSION

We propose *SpikeGen*, the first latent generative framework specifically designed for decoupled visual representation processing. We successfully pre-trained the model on more than one million images and demonstrated its broad applicability across diverse tasks, covering all major domains of joint spike stream and RGB visual processing. We believe this can serve as a foundation for advancing latent generative modeling in neuromorphic vision, closely mirroring the human capacity to integrate perception and imagination for multimodal visual processing in dynamic environments.

## 5   ETHICS STATEMENT

This paper presents work whose goal is to advance the field of Machine Learning. There are many potential societal consequences of our work, none which we feel must be specifically highlighted here.

## 6   REPRODUCIBILITY STATEMENT

To ensure the reproducibility of the computational experiments, we have provided the code in `https://github.com/zhenwuweihe/SpikeGen`. Detailed information regarding the selected baselines, data usage, and codebases is presented in the Appendix.

## 7   ACKNOWLEDGEMENTS

G.D. was supported by the National Natural Science Foundation of China (W2442028).

R.Z. was supported by the National Natural Science Foundation of China (625B2090).

S.Z. was supported by the National Natural Science Foundation of China (62476011), and by the Beijing Natural Science Foundation (L252060).

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

## A  Experiment Setups

### A.1  Baselines

**Conditional Video Deblurring**

- Jin et al. (Jin et al., 2018) proposed LEVS, which tackled the extraction of a video sequence from a single motion-blurred image by proposing a deep learning scheme that sequentially reconstructs pairs of frames using loss functions invariant to their temporal order.

- Zhang et al. (Zhang et al., 2021b)proposed MotionETR, focusing on recovering dense, potentially non-linear exposure trajectories from a motion-blurred image by proposing a novel motion offset estimation framework to model pixel-wise displacements over time.

- Zhong et al. (Zhong et al., 2023) addressed real-world motion recovery from blur (joint deblurring and interpolation) by introducing a Blur Interpolation Transformer (BiT) that utilizes multi-scale residual Swin Transformer blocks and temporal supervision strategies.

- Sun et al. (Sun et al., 2023) worked on event-based video frame interpolation with the capability of ad-hoc deblurring for both sharp and blurry input frames, using a bidirectional recurrent network (REFID) that adaptively fuses image and event information.

- Xu et al. (Xu et al., 2021) tackled motion deblurring using real-world event data through a self-supervised learning framework (RED-Net) that leverages event-predicted optical flow for blur and photometric consistency constraints.

- Chen and Yu (Chen & Yu, 2024) addressed event-based motion deblurring by proposing a Two-stage Residual-based Motion Deblurring (TRMD) framework, which first estimates an intensity residual sequence from events and then uses it to reconstruct sharp frames from the blurry image.

- Chen et al. (Chen et al., 2023b) aimed at enhancing motion deblurring in high-speed scenes by proposing SpkDeblurNet, a model that leverages spike streams as auxiliary cues and employs content-aware motion magnitude attention and transposed cross-attention fusion for RGB-spike data integration.

- Chen et al. (Chen et al., 2024) (SpikeReveal) focused on unlocking temporal sequences of sharp images from real blurry inputs assisted by spike streams, using a self-supervised spike-guided deblurring model (S-SDM) that explores theoretical relationships between the modalities.

**Dense Frame Reconstruction**

- Zhu et al. Zhu et al. (2019) addressed visual texture reconstruction from spike streams by proposing a retina-inspired sampling method (TFP and TFI) and associated spike stream decoding techniques.

- Zheng et al.  Zheng et al. (2021) tackled high-speed image reconstruction from spiking cameras by introducing novel models (TFSTP and TFMDSTP) based on the short-term plasticity (STP) mechanism of the brain.

- Zhang et al. Zhang et al. (2023a) focused on learning effective temporal-ordered representations for spike streams by proposing a Wavelet-Guided Spike Enhancing (WGSE) paradigm that utilizes discrete wavelet transforms.

- Zhao et al. Zhao et al. (2021) developed Spk2ImgNet, a spike-to-image neural network, to learn the reconstruction of dynamic scenes from continuous spike streams generated by spiking cameras.

- Zhao et al. Zhao et al. (2023) proposed SSIR to addressed spike camera image reconstruction by employing deep Spiking Neural Networks (SNNs), aiming for comparable performance to state-of-the-art methods but with lower computational costs.

- Fan et al. Fan et al. (2024) tackled efficient image reconstruction for spiking cameras by proposing a spatio-temporal interactive reconstruction network (STIR) that jointly performs inter-frame feature alignment and intra-frame feature filtering in a coarse-to-fine manner.

- Ercan et al. Ercan et al. (2024) aimed to improve event-based video reconstruction by proposing HyperE2VID, a dynamic neural network architecture that uses hypernetworks to generate per-pixel adaptive filters guided by a context fusion module.

- Weng et al. Weng et al. (2021) addressed event-based video reconstruction by proposing ET-Net, a hybrid CNN-Transformer framework designed to leverage both fine local information from CNNs and global context from Transformers.

**Novel-View Synthesis**

- Kerbl et al. Kerbl et al. (2023) introduced 3D Gaussian Splatting for real-time, high-quality radiance field rendering by representing scenes as a collection of 3D Gaussians optimized from sparse points and rasterized efficiently.

- Ma et al. Ma et al. (2022) addressed the reconstruction of Neural Radiance Fields (NeRFs) from blurry images by proposing Deblur-NeRF, which simulates the blurring process using a deformable sparse kernel module within an analysis-by-synthesis framework.

- Chen and Liu Chen & Liu (2024) tackled 3D scene reconstruction using Gaussian Splatting from camera motion-blurred images (Deblur-GS) by jointly optimizing 3D Gaussian parameters and the camera's motion trajectory during exposure time.

- Guo et al. Guo et al. (2024) proposed Spike-NeRF, the first Neural Radiance Field based method for 3D reconstruction and novel view synthesis of high-speed scenes using continuous spike streams from a moving spike camera, incorporating spike masks and a specialized loss function.

- Guo et al. Guo et al. (2025) developed SpikeGS to reconstruct 3D scenes from Bayer-pattern spike streams captured by a fast-moving bio-inspired camera by integrating spike data into the 3D Gaussian Splatting pipeline using accumulation rasterization and interval supervision.

## A.2 DATASETS

**Pre-Training**   We utilize the complete training set from ImageNet (Deng et al., 2009) to synthesize spike-RGB pairs. ImageNet comprises 1,000 categories, with each category containing approximately 1,000 RGB images, resulting in a total of roughly 1 million images. For each clear RGB image, we transform it into a blurred image and an 8-frame spike stream. The RGB blur kernel is configured as $40 \times 40$, and the coverage rate of the spike stream is set to 0.1. The Dataset and Dataloader code are provided in the supplementary materials.

**Conditional Video Deblurring**   To facilitate a quantitative analysis of our spike-assisted motion deblurring performance, we constructed a synthetic dataset leveraging the commonly adopted GO-PRO (Nah et al., 2017) dataset. The process commenced with the application of the XVFI (Sim et al., 2021) interpolation algorithm, which served to augment the video data by generating seven additional intermediate frames between every pair of successive sharp images. For the creation of spike streams that faithfully mimic real-world sensor data, the interpolated video sequences were first spatially downscaled to a $320 \times 180$ resolution, followed by simulation using a dedicated spike simulator (Zhao et al., 2022). To replicate the effects of real-world motion blur, each blurry input frame was then synthesized through the averaging of 97 consecutive frames from these upsampled video sequences.

**Dense Frame Reconstruction**   Following STIR (Fan et al., 2024), the SREDS dataset, a recently introduced collection that was synthesized from the established REDS (Xu et al., 2021) dataset. This dataset is organized into 240 distinct scenes for training and 30 separate scenes for testing. Each scene consists of 24 sequential frames, with every frame being accompanied by a corresponding spike stream (N=20) that is centrally aligned to it. The spatial resolution for this data is $1280 \times 720$ pixels. In preparation for training, we crop each scene into non-overlapping patches of 96×96 pixels, which yields a total of 21,840 patches. To evaluate our model's effectiveness, we employ real-world captured data. This validation set includes the "momVidarReal2021" (Zhu et al., 2020) publicly accessible dataset, which has a resolution of $400 \times 250$ and features significant object and camera motion at high speeds (this dataset has also been previously employed in other studies).

**Novel-View Synthesis**   We use the same dataset used in SpikeGS (Guo et al., 2025). It first contains the Blender dataset presented by NeRF (Mildenhall et al., 2021), this dataset comprises path-traced images of eight objects characterized by intricate geometry and realistic non-Lambertian materials. For six of these objects, viewpoints were sampled across the upper hemisphere, while the other two were rendered from viewpoints covering a full sphere. For every scene in this dataset, 100 views are rendered to serve as input and 200 views for testing purposes, all at an $800 \times 800$ pixel resolution. The second one is the Blender dataset presented by DeblurNeRF (Ma et al., 2022). This dataset synthesized five distinct scenes, with multi-view cameras manually positioned to simulate realistic data capture conditions. To create images with camera motion blur, it first introduced random perturbations to the camera poses and then performed linear interpolation between the original and perturbed poses for each view; images rendered from these interpolated poses were subsequently blended in linear RGB space to form the final blurry images.

## A.3   METRICS

Several metrics are commonly used to evaluate the quality of images, differing in whether they require a reference (Full-Reference, FR) or operate without one (No-Reference, NR).

**Peak Signal-to-Noise Ratio (PSNR)** is an FR metric that measures image fidelity based on the Mean Squared Error (MSE) between a reference image $I$ and a processed image $K$, both of size $m \times n$. A higher PSNR generally indicates better reconstruction quality. It is defined as:

$$MSE = \frac{1}{m \cdot n} \sum_{i=0}^{m-1} \sum_{j=0}^{n-1} [I(i,j) - K(i,j)]^2 \tag{4}$$

$$PSNR = 10 \cdot \log_{10} \left( \frac{MAX_I^2}{MSE} \right) \tag{5}$$

where $MAX_I$ is the maximum possible pixel value of the image (e.g., 255 for 8-bit images).

**Structural Similarity Index Measure (SSIM)** is another FR metric designed to better align with human perception of image quality by considering changes in structural information, luminance, and contrast. For two image windows $x$ and $y$, SSIM is calculated as:

$$SSIM(x,y) = \frac{(2\mu_x\mu_y + c_1)(2\sigma_{xy} + c_2)}{(\mu_x^2 + \mu_y^2 + c_1)(\sigma_x^2 + \sigma_y^2 + c_2)} \tag{6}$$

where $\mu_x, \mu_y$ are the local means, $\sigma_x^2, \sigma_y^2$ are the local variances, $\sigma_{xy}$ is the local covariance, and $c_1, c_2$ are small constants to stabilize the division. The overall SSIM is typically the average of local SSIM values. Higher SSIM values (closer to 1) indicate greater similarity.

**Learned Perceptual Image Patch Similarity (LPIPS)**, proposed by Zhang et al. (Zhang et al., 2018), is an FR metric that aims to better reflect human perceptual judgments by comparing deep features extracted from images using pre-trained convolutional neural networks. The distance $d$ between a reference image $x_0$ and a distorted image $x$ is computed by summing the L2 distances of their feature activations $\hat{y}^l, \hat{y}_0^l$ (normalized and channel-wise scaled by $w_l$) across different layers $l$ of a network:

$$d(x, x_0) = \sum_l \frac{1}{H_l W_l} \sum_{h,w} ||w_l \odot (\hat{y}_{hw}^l - \hat{y}_{0hw}^l)||_2^2 \tag{7}$$

Lower LPIPS scores indicate higher perceptual similarity.

**Natural Image Quality Evaluator (NIQE)**, introduced by Mittal et al. (Mittal et al., 2012b), is an NR (blind) IQA metric. It operates by constructing a model based on natural scene statistics (NSS) extracted from a corpus of pristine natural images. The quality of a test image is then measured by the distance between the NSS features extracted from it and the pristine model. NIQE does not require training on human opinion scores of distorted images. Lower NIQE scores suggest better perceptual quality, closer to natural image statistics. The core idea involves fitting a multivariate Gaussian model to features derived from patches of natural images and then measuring the deviation of test image patch features from this model.

**Blind/Referenceless Image Spatial Quality Evaluator (BRISQUE)**, developed by Mittal et al. (Mittal et al., 2012a), is another NR IQA model that operates in the spatial domain. It quantifies the loss

of "naturalness" in an image by analyzing deviations in its locally normalized luminance coefficients (specifically, Mean Subtracted Contrast Normalized - MSCN coefficients) from statistical models of natural images. Features derived from the MSCN coefficient distributions are used to train a Support Vector Regressor (SVR) to predict a subjective quality score. Lower BRISQUE scores generally indicate better quality.

## A.4 CODE-BASES

**Pre-Training**  Our code is based on the implementation of MAR (Li et al., 2024).

Link:`https://github.com/LTH14/mar`

**Conditional Video Deblurring**  Our code is based on the implementation of S-SDM (Chen et al., 2024).

Link:`https://github.com/chenkang455/S-SDM`

**Dense Frame Reconstruction**  Our code is based on the implementation of STIR (Fan et al., 2024).

Link:`https://github.com/GitCVfb/STIR`

**Novel-View Synthesis**  Our code is based on the implementation of SpikeGS (Guo et al., 2025).

Link:`https://github.com/yijiaguo02/SpikeGS`

# B  TRAINING CONFIGURATIONS

All experiments were carried out on eight A800 GPUs, utilizing AdamW as the optimizer with a weight decay rate of 0.05. The training schedule was configured to include a warm-up phase (spanning 10% of the total epochs), during which the learning rate increased from 1e-8 to the target value, followed by a cosine decay to a minimum learning rate of 1e-6.

## B.1  MODEL

All the results are conducted with the same model hyperparameters listed in Table 4

Table 4: **Hyperparameters for SpikeGen model**

| Parameter | Value |
|---|---|
| `encoder_embed_dim` | 768 |
| `encoder_depth` | 12 |
| `encoder_num_heads` | 12 |
| `decoder_embed_dim` | 768 |
| `decoder_depth` | 12 |
| `decoder_num_heads` | 12 |
| `mlp_ratio` | 4 |
| `norm_layer` | `partial(nn.LayerNorm, eps=`$10^{-6}$`)` |
| `max_position_embeddings` | 2048 |
| `RoPE base` | 10000 |

Table 5: Detailed Specifications of the S3 Encoder (Spatial-Temporal Separable Spike Encoder)

| Module Category | Subcomponent Name | Function Description | Key Parameters | Input Shape | Output Shape |
|---|---|---|---|---|---|
| Spatio-Temporal Feature Extraction Module | Initial Feature Extraction Block | Perform preliminary feature encoding on raw spike streams | Conv3d: 1→32 channels, 3×3×3 kernel, stride=1, padding=1; LeakyReLU(0.2); InstanceNorm3d | $[B, 1, T, H, W]$ | $[B, 32, T, H, W]$ |
| | Downsampling Block 1 | First spatio-temporal downsampling, increase channel depth | Conv3d: 32→64 channels, 3×3×3 kernel, stride=2, padding=1; LeakyReLU(0.2); InstanceNorm3d | $[B, 32, T, H, W]$ | $[B, 64, T/2, H/2, W/2]$ |
| | Downsampling Block 2 | Second spatio-temporal downsampling, enhance feature expression | Conv3d: 64→128 channels, 3×3×3 kernel, stride=2, padding=1; LeakyReLU(0.2); InstanceNorm3d | $[B, 64, T/2, H/2, W/2]$ | $[B, 128, T/4, H/4, W/4]$ |
| | Downsampling Block 3 | Third spatio-temporal downsampling, achieve 512× downsampling | Conv3d: 128→256 channels, 3×3×3 kernel, stride=2, padding=1; LeakyReLU(0.2); InstanceNorm3d | $[B, 128, T/4, H/4, W/4]$ | $[B, 256, T/8, H/8, W/8]$ |
| | Channel Adjustment Conv | Adjust feature channels to target dimension (512) | Conv3d: 256→512 channels, 1×1×1 kernel, stride=1; LeakyReLU(0.2) | $[B, 256, T/8, H/8, W/8]$ | $[B, 512, T/8, H/8, W/8]$ |
| Temporal Fusion Module | Temporal Attention Generation | Generate temporal weights to highlight critical temporal information | 2-layer Conv3d: 512→512 channels, 1×1×1 kernel; LeakyReLU(0.2); Sigmoid activation | $[B, 512, T/8, H/8, W/8]$ | $[B, 512, T/8, H/8, W/8]$ |
| | Temporal Weight Application | Apply temporal attention to spatio-temporal features, enhance valid info | Element-wise multiplication (features × temporal weights) | $[B, 512, T/8, H/8, W/8]$ | $[B, 512, T/8, H/8, W/8]$ |
| | Temporal Dimension Summation | Fuse features along the temporal axis, compress temporal dimension | Summation along the T-axis (dim=2) | $[B, 512, T/8, H/8, W/8]$ | $[B, 512, H/8, W/8]$ |
| | Feature Refinement | 2D Conv + normalization to optimize spatial feature consistency | Conv2d: 512→512 channels, 3×3 kernel, padding=1; LayerNorm; LeakyReLU(0.2) | $[B, 512, H/8, W/8]$ | $[B, 512, H/8, W/8]$ |

## B.2  TRAINING

We listed the hyperparameters for pre-training and fine-tuning in Table 6, covering the necessary hyperparameters which is not explicitly outlined in our codebases.

Table 6: **Training Hyperparameters**

| Parameter | ImageNet | GOPRO | SREDS | Blender |
|---|---|---|---|---|
| Learning Rate | $1.00 \times 10^{-4}$ | $1.00 \times 10^{-4}$ | $1.00 \times 10^{-4}$ | $1.00 \times 10^{-5}$ |
| Epoch | 10 | 100 | 150 | 500 |
| Batch Size | 64 | 32 | 64 | 32 |
| Image Resolution | $256 \times 256$ | $768 \times 384$ | $96 \times 96$ | $800 \times 800$ |
| Spike Frame # | 8 | 8 | 24 | 8 |

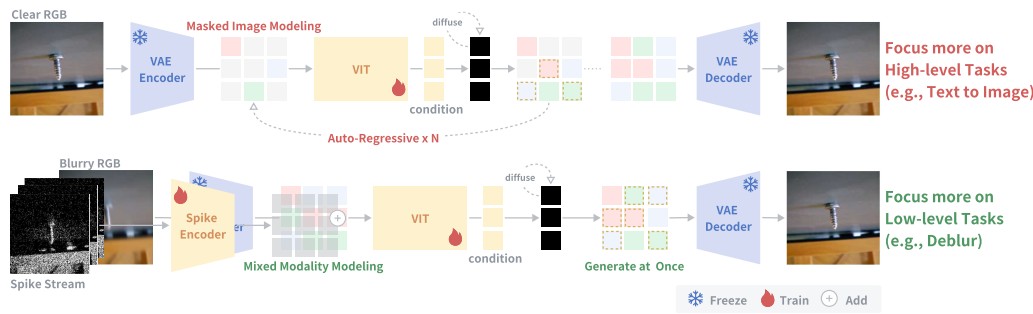

Figure 6: **Modified SpikeGen Pipeline for Low-level Tasks**

## C ADDITIONAL RESULTS

### C.1 EFFICIENT LATENT PROCESSING WITHOUT AUTO-REGRESSION

As outlined in Section 3, our model emphasizes the integration and restoration of existing visual tokens rather than generating new tokens from void. In light of this focus, and to further enhance computational efficiency, we eliminated the autoregressive component that was present in the original MAR framework. It is important to note that this architectural modification does not compromise the model's performance, as demonstrated in Table 7.

Table 7: **Efficiency Comparison: SpikeGen vs. Diffusion Baselines**

| Dataset: ImageNet | Device: 1×NVIDIA A100 80G | Resolution: 256×256 | | | | |
|---|---|---|---|---|
| Backbone | time/256 images | PSNR | SSIM | LPIPS |
| ***DiT-B*** ($100 \times$ ViT steps) $\times$ 256 tokens | 2min19s | 18.11 | 0.59 | 0.28 |
| ***MAR-B*** ($64 \times$ ViT steps + $100 \times$ MLP steps) $\times$ gradually to 256 tokens | 1min32s | 18.79 | 0.65 | 0.21 |
| ***SpikeGen*** ($1 \times$ **ViT steps** + $100 \times$ **MLP steps**) $\times$ **256 tokens** | 2.7s | 19.08 | 0.62 | 0.19 |

### C.2 GENERALIZATION ON IMAGENET

We present extra visualization results as an extended support for the discussion in Section 4.1.

### C.3 CONDITIONAL VIDEO DEBLURRING

As the conclusion elaborated in detail in the main text, introducing the spike modality can greatly alleviate the ambiguity of deblurring images, thus avoiding falling into the "sharpness trap" of single RGB modality deblurring. In Figure 8, we demonstrate that this property can be successfully transferred to downstream tasks. We can observe that, regardless of the sparsity of the introduced spikes (controlled by $V_{th}$), the final deblurring results are superior to those relying on a single RGB modality. Moreover, as $V_{th}$ decreases (indicating denser spike information), this improvement becomes more pronounced. From the visualization results, we can also see that although the overall texture is similar, introducing the spike modality can significantly enhance the restoration details.

### C.4 DENSE FRAME RECONSTRUCTION

We mentioned in the main text that using the results of TFP as pseudo-dense frames can significantly improve the performance of the model, and we further demonstrated this in Table 8. When relying solely on spike streams, the model has difficulty directly fusing spatially sparse binary spike streams. Especially when the number of spike frames is insufficient. However, simply increasing the number of spike frames means that the model needs to multiply the inference consumption. Therefore, using the results of TFP as pseudo-dense frames is a good balancing operation. When using 1/4 of the spike frames (we use 16 frames while STIR (Fan et al., 2024) uses 64 frames), we have improved

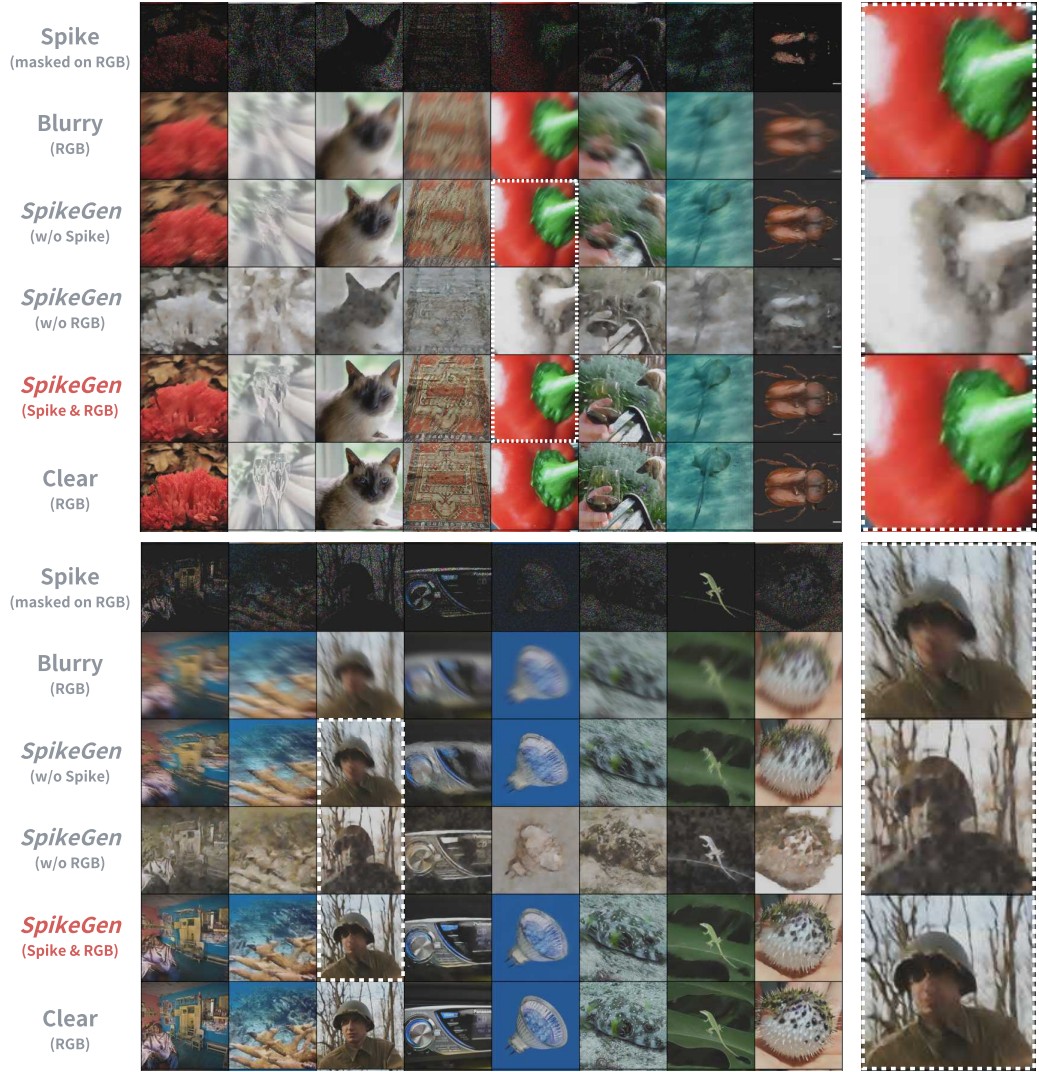

Figure 7: **Extra Experiments: In-domain Generalization for Conditional Image Deblurring**

Table 8: **Ablation Study: Pseudo Dense Frame Control on SpikeGen**

| Methods | Dataset: *SREDS* (Synthetic) (Zhao et al., 2023) | | | | | Dataset: *momVidar2021* (Real) (Zhu et al., 2020) | |
|---|---|---|---|---|---|---|---|
| | PSNR ↑ | SSIM ↑ | LPIPS ↓ | NIQE ↓ | BRISQUE ↓ | NIQE ↓ | BRISQUE ↓ |
| *TFP* (ICME19) (Zhu et al., 2019) | 25.35 | 0.69 | 0.26 | 5.97 | 43.07 | 9.34 | 45.20 |
| *TFI* (ICME19) (Zhu et al., 2019) | 18.50 | 0.64 | 0.26 | 4.52 | 44.93 | 10.10 | 58.31 |
| *SpikeGen* (Spike) | 33.62 | 0.94 | 0.05 | 3.11 | 15.37 | 5.70 | 22.03 |
| *SpikeGen* (TFI&Spike) | 37.19 | 0.96 | 0.04 | 3.24 | 15.92 | 5.60 | 16.76 |
| *SpikeGen* (TFP&Spike) | **39.25** | **0.98** | **0.01** | **2.83** | **14.99** | **5.33** | **15.97** |

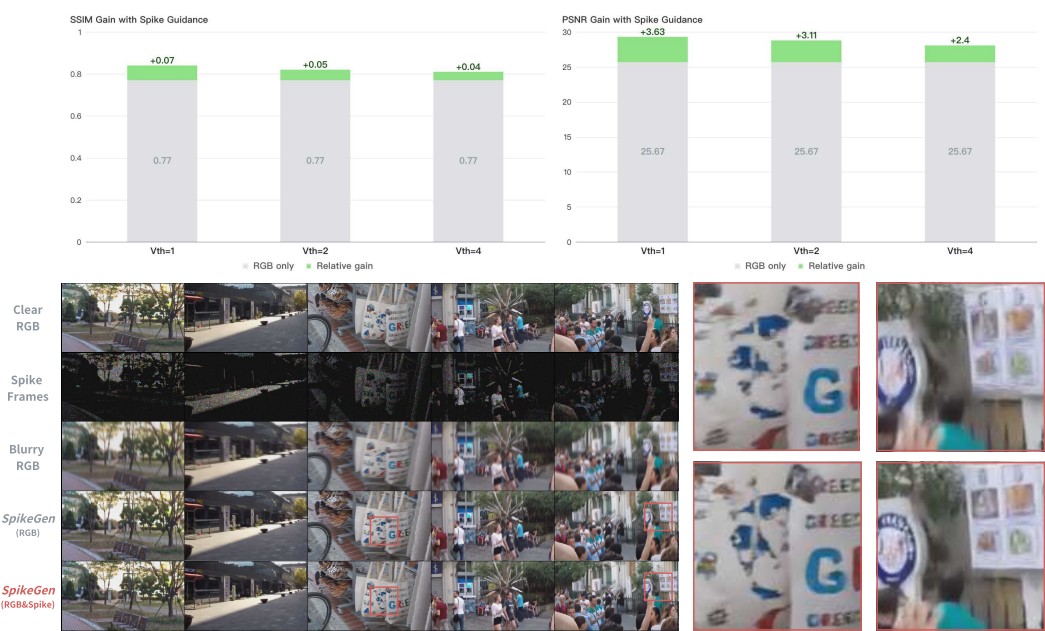

Figure 8: **Ablation Study: Modality Control on SpikeGen with GOPRO Dataset**

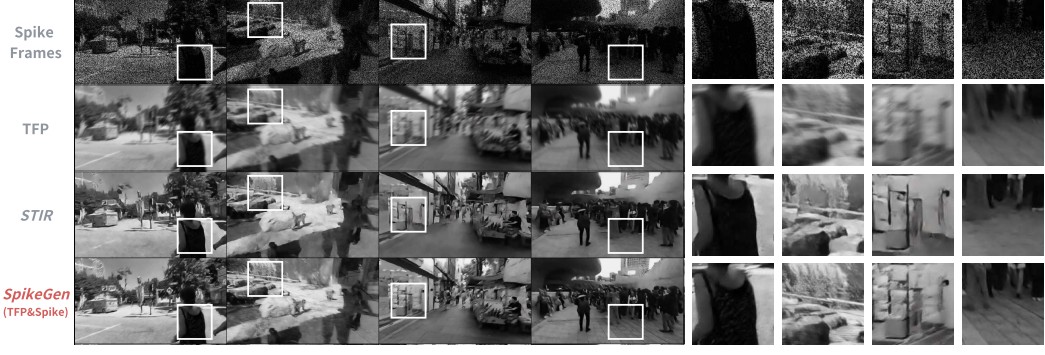

Figure 9: **Ablation Study: Stress Test under Limited Spike Information.**

Table 9: **Extra Experiments: Comparison to Other Two-Stage Methods**

| | | **Objects** (Mildenhall et al., 2021) | | | **Out-Door** (Ma et al., 2022) | | | **Average** | | |
|---|---|---|---|---|---|---|---|---|---|---|
| Methods | Dual Modality | PSNR ↑ | SSIM ↑ | LPIPS ↓ | PSNR ↑ | SSIM ↑ | LPIPS ↓ | PSNR ↑ | SSIM ↑ | LPIPS ↓ |
| *3DGS* (Clear) (Kerbl et al., 2023) | ✗ | 33.31 | 0.96 | 0.05 | 30.27 | 0.91 | 0.10 | 31.79 | 0.94 | 0.07 |
| *3DGS* (Blur) (Kerbl et al., 2023) | ✗ | 26.95 | 0.88 | 0.12 | 23.38 | 0.69 | 0.45 | 25.16 | 0.78 | 0.28 |
| *MPR* (3DGS) (CVPR21) (Zamir et al., 2021) | ✗ | 28.73 | 0.90 | 0.12 | 27.01 | 0.73 | 0.40 | 27.87 | 0.81 | 0.19 |
| *TFI* (3DGS) (CVPR21) (Zheng et al., 2021) | ✓ | 29.97 | 0.92 | 0.11 | 26.96 | 0.82 | 0.25 | 29.92 | 0.84 | 0.14 |
| *TFP* (3DGS) (ICME19) (Zhu et al., 2019) | ✓ | 30.11 | 0.92 | 0.09 | 27.51 | 0.76 | 0.36 | 29.14 | 0.80 | 0.22 |
| *EDI* (3DGS) (CVPR19) (Pan et al., 2019) | ✓ | 29.45 | 0.92 | 0.11 | 27.88 | 0.79 | 0.32 | 29.73 | 0.84 | 0.12 |
| *STIR* (3DGS) (NeurIPS24) (Fan et al., 2024) | ✓ | 29.01 | 0.92 | 0.12 | 28.01 | 0.83 | 0.20 | 29.45 | 0.87 | 0.16 |
| ***SpikeGen*** (3DGS) | ✓ | **31.81** | **0.94** | **0.07** | **28.26** | **0.89** | **0.13** | **30.04** | **0.92** | **0.10** |

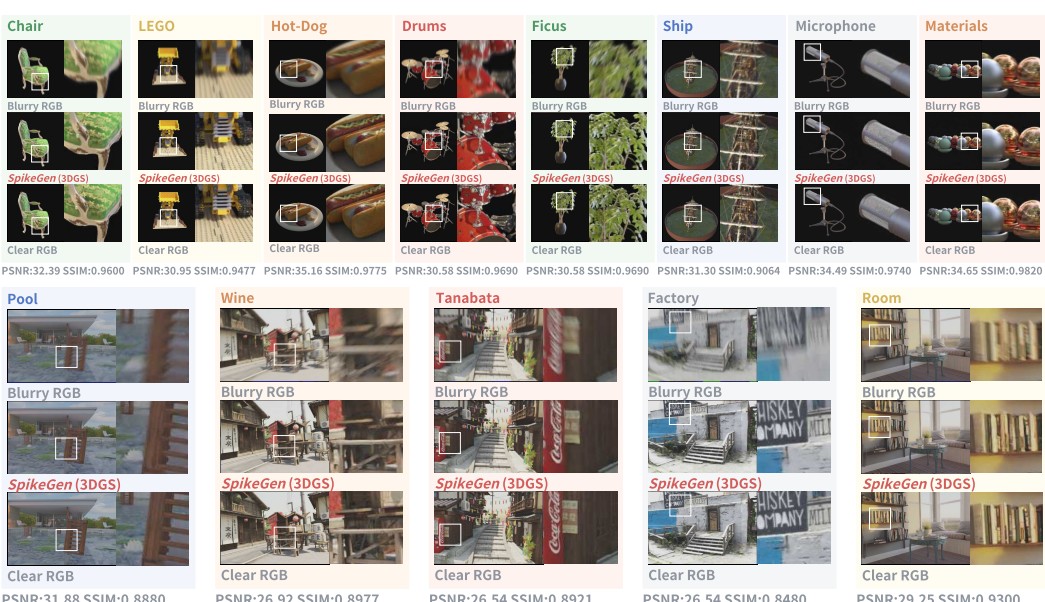

Figure 10: **Extra Experiments: Category Evaluation of SpikeGen on Blender Dataset**

the performance of SpikeGen to the state-of-the-art level, even though the focus of our work is not on single-modal tasks. To further verify our conclusion, we conducted a more extreme comparison. In Figure 9, we further reduced the number of spike frames to 8. In this case, STIR is unable to restore the spatial details of the object, while our method can still maintain a certain level of detail restoration, such as the texture of the floor tiles.

## C.5 Novel-View Synthesis

We mentioned in the main text that our method is not an end-to-end model similar to SpikeGS Guo et al. (2025). However, we believe that this two-stage method also has potential advantages. For example, it can better leverage pre-trained priors and decouple the complexity of optimizing texture and perceptual fidelity. Therefore, we also compared the effects of using other spike processors for two-stage reconstruction. As shown in Table 9, although this two-stage method is feasible, it is not easy to achieve stable excellence. In the Object data, since the background (pure black) and objects are relatively simple, the other two-stage methods also perform well. However, in outdoor scenes, due to the complex background and tiny components, the performance of the other two-stage methods is greatly affected, while SpikeGen can still maintain a relatively stable performance. We further presented all the test contents on the two datasets and the specific metrics of our method in Figure 10.

## D    THE USE OF LARGE LANGUAGE MODELS (LLMS)

This paper solely employs Large Language Models to refine written content, encompassing grammar correction, tone adjustment, and formatting.

