# OpenReview forum: "SpikeGen: Decoupled “Rods and Cones” Visual Representation Processing with Latent Generative Framework"
_ICLR.cc/2026/Conference — ICLR 2026 Poster_

### Official Review · Reviewer_CRQ3 · 2025-10-29

**Soundness:** 3
**Presentation:** 3
**Contribution:** 4
**Rating:** 6
**Confidence:** 5

**Summary:**

Mimicking the decoupling of color and light intensity perception of human into cone and rod cells, the authors proposed SpikeGen, which outperforms previous state-of-the-art methods in multiple spike-RGB tasks: 1) conditional image/video deblurring, 2) dense frame reconstruction and 3) scene novel-view sythesis.

SpikeGen follows the general setup of a Masked Auto-regressive mode (MAR) pretrained with diffusion loss. It uses a frozen VAE encoder and a trainable spike encoder to encode blurry RGB input and spike stream into latent representations, respectively.

SpikeGen has two training stages, a pretraining stage and a finetuning stage. During the pretraining, the loss used is the per-token diffusion loss between the faded clear RGB latent and the latent predicted by the ViT followed by a compact MLP. The spike stream loss is used during the finetuning stage.

The experiment results reported in the paper show that SpikeGen beats all other benchmarks on all three tasks.

**Strengths:**

1. The empirical results reported by the authors on the three tasks show the superior performances of SpikeGen over all previous methods compared in the paper. The superiority of the proposed method and training pipeline is consistent across various tasks, metrics and threshold settings.

2. The figures presented in the paper are clear, well-organized, and informative, especially Figure 2. It effectively and clearly conveys the overall training pipeline while also giving details on the architecture of the Spatial–Temporal Separable Spike Encoder. This visualization greatly improves the reader’s understanding of the specific modifications made to the original MAR pipeline.

3. The adoption of the latent diffusion training for a dual modality spike-RGB self-supervised training method is both innovative and effective. The introduction of a random gamma parameter during training enables controllable modality dominance, allowing adaptation to different downstream tasks with minimal effort.

**Weaknesses:**

1. Although the authors briefly mention at line 475 that SpikeGen’s novel-view synthesis pipeline is two-stage, the main text provides insufficient explanation. It remains unclear to me 1) what the two stages are specifically (deblurring + vanilla 3DGS?),  2) how they differ from other benchmarks reported in Table 3, and 3) how time efficient (or inefficient) is SpikeGen compared to other single-stage methods. A more detailed description and comparison would greatly improve the paper’s completeness.

2. Table 9 in the appendix provides comparisons with other two-stage NVS methods. However, it only includes two relatively old, training-free approaches. The authors are encouraged to include other two-stage methods, e.g. those listed in Table 2, to provide a more complete and up-to-date comparison.

3. An ablation study replacing MAR with a regular conditional DiT diffusion model would provide a much stronger basis for SpikeGen’s architectural design choices.

4. Similarly, an ablation on design choices of the S3 encoder would also further validate the architecture's effectiveness. Nevertheless, considering the page limit, this omission is understandable and does not substantially affect my overall evaluation.

5. At lines 69–70, the phrase “By reviewing current studies” appears to be a typo or editing error.

**Questions:**

1. Have you tried to linearly interpolate gamma values for testing? How does gamma values affect performances?

2. Please see the weaknesses section for additional questions and suggestions

---

> ### Author Response · Authors · 2025-11-17
> **Reply for Reviewer CRQ3 [W1,2,3,4]**
>
> We sincerely thank the reviewer for their strong support and positive feedback. We have read the comments carefully and are grateful for the professional and insightful analysis. Your feedback is invaluable for further improving our paper. We will now provide a point-to-point response.
> ## Weakness 1 (Details on the two-stage NVS pipeline):
> You are correct in your understanding. Our approach for `Task 3` (NVS) is a two-stage pipeline:
> 1. We first deblur all multi-view input images using our pre-trained SpikeGen.
> 2. We then feed these clear images into a standard 3DGS model for reconstruction.
>
> This approach differs from the single-stage baselines presented in `Table 3` (e.g., DeblurNeRF, SpikeGS), which adapt the NeRF/3DGS framework to implicitly learn static scene structures from blurred inputs without generating explicit deblurred 2D images. As noted, both paradigms involve trade-offs. Single-stage methods typically exhibit a higher upper bound, as they learn scene-specific deblurring capabilities; however, this comes at the cost of training instability due to the challenging multi-objective optimization process that jointly optimizes deblurring and 3D reconstruction. In contrast, two-stage approaches have a stronger lower bound and offer greater controllability, as the final 3D quality directly dependent on the explicitly deblurred 2D inputs. Our objective of this experiment is to demonstrate the broad downstream applicability of SpikeGen, we adopt a straightforward modification by applying 3DGS in the second stage.
>
> Regarding your third point on efficiency, our **two-stage method is actually efficient** compared to single-stage baselines, which add computational overhead to the already expensive 3DGS/NeRF training. The cost for the first stage of SpikeGen on **all images** in the Blender dataset is around 1 minute on one A100 (80g) GPU. This is a negligible cost compared to the 3DGS training (e.g., ~6 minutes/scene) **[1]**.
> ## Weakness 2 (Additional two-stage baselines):
> We are happy to add new results. Following the baseline setups in E2NeRF **[2]**, we have added new two-stage comparisons using different image debluring preprocessor, they are MPR **[3]** (RGB-based) and EDI **[4]** (event-based).
>
> **Dataset: Blender Objects**
> | Methods | PSNR | SSIM |
> | :--- | :---: | :---: |
> | 3DGS (Clear) | 33.31 | 0.96 |
> | 3DGS (Blur) | 26.95 | 0.88 |
> | TFI+3DGS | 29.97 | 0.92 |
> | TFP+3DGS | 30.11 | 0.92 |
> | MPR+3DGS | 28.73 | 0.90 |
> | EDI+3DGS | 29.45 | 0.92 |
> | **SpikeGen+3DGS** | **31.81** | **0.94** |
> ## Weakness 3 (MAR vs. DiT):
> This is a great question about our architectural choice. In fact, we did consider DiT-based backbones in the early stages.
> We found that a standard conditional DiT is "bulky" and inefficient for our task. A DiT requires applying the **entire ViT backbone** for multiple diffusion steps on all tokens. In contrast, MAR applies the ViT as an **auto-regressive (AR) encoder**, which generate the tokens gradually, with the diffusion process handled by a **lightweight MLP head**.
>
> This efficiency gap is significantly widened by our finding in `Table 7`, which shows that our model can achieve SOTA performance **even without auto-regressive** steps. This further separates our model from DiT in both efficiency and performance.
>
> We hypothesize this is due to the fundamental difference in tasks. Original DiT/MAR excel at **high-level (e.g., text) to low-level (image) generation**, where iterative refinement is essential. Ours is a **low-level (image-to-image) task**, focused on integrating existing information based on semantic priors. In this regime, a long refinement process yields marginal returns and may even corrupt high-quality information already present in the input.
>
> **Device: 1\*NVIDIA A100 80G | Resolution: 256\*256**
> | Backbone | time/256 images | Cost List | Test PSNR (ImageNet) |
> | :--- | :---: | :--- | :---: |
> | DiT-B | 2min19s | (100\*ViT Forward)\*256 tokens | 18.11 |
> | MAR-B | 1min32s | (64\*ViT+100\*MLP Forward)\*gradually to 256 tokens | 18.79 |
> | **SpikeGen** | **2.7s** | **(1\*ViT+100\*MLP Forward)\*256 tokens** | **19.08** |
> ## Weakness 4 (Choice of S3 Encoder):
> We appreciate your understanding of the page limits. Our design for the S3 encoder was guided by two principles:
> 1. **Motivation**: Following our bio-inspiration, we wanted a lightweight encoder (the "eyes") that only handles encoding, while the central ViT (the "cortex") does the complex work of information integration. This allows us to leverage powerful, pre-trained generative backbones (like MAR) to provide semantic priors and avoid the "sharpness trap".
> 2. **Efficiency**: A complex spike encoder would require a weaker ViT backbone to maintain the same computational budget.
>
> Therefore, our S3 encoder choose general components (3D/2D conv) but remains spike-specific designs (spatial-temporal separable  + temporal weighted attention) to balance this trade-off.

---

> ### Author Response · Authors · 2025-11-17
> **Reply for Reviewer CRQ3 [W5,Q1,2]**
>
> ## Weakness 5 (Typo):
> We would correct this typo in the revised PDF. Thank you for your careful reading.
> ## Question 1 (Testing with $\gamma$ interpolation):
> Yes! This is an interesting function we support. Through pre-training, we can change the behavior pattern of the model by adjusting the fusion degree of modalities. Please note that in `Figure 2, 3` (main text), `7, 8` (appendix), we demonstrate the different visual effects that can be obtained by regulating this mixing ratio. Moreover, `Table 8` and `Figure 8` (appendix) show the quantitative results when controlling the proportion of different modalities. To summarize briefly, the mixed modality generally achieves better average performance, which is reflected in most of our quantitative results. However, in some specific situations, such as the overexposure scenario of an ordinary RGB camera, better performance can be obtained by increasing the proportion of the spike modality.
> ## Question 2:
> Please refer to the reply above
> ## Reference
> [1] Kulhanek, Jonas, and Torsten Sattler. "Nerfbaselines: Consistent and reproducible evaluation of novel view synthesis methods." arXiv preprint arXiv:2406.17345 (2024).
>
> [2] Qi, Yunshan, et al. "E2nerf: Event enhanced neural radiance fields from blurry images." Proceedings of the IEEE/CVF International Conference on Computer Vision. 2023.
>
> [3] Zamir, Syed Waqas, et al. "Multi-stage progressive image restoration." Proceedings of the IEEE/CVF conference on computer vision and pattern recognition. 2021.
>
> [4] Pan, Liyuan, et al. "Bringing a blurry frame alive at high frame-rate with an event camera." Proceedings of the IEEE/CVF conference on computer vision and pattern recognition. 2019.

---

> > ### Comment · Reviewer_CRQ3 · 2025-11-17
> >
> > I appreciate the authors’ detailed and thoughtful response to my comments and questions.
> >
> > All of my concerns have been satisfactorily addressed except for one remaining issue.
> >
> > Regarding Weakness 2, the newly added methods appear to be new additions that are not mentioned in the main text. I am curious why spike-based methods such as STIR were not included in the comparison for the two-stage NVS experiments. Since STIR ranks as a close second to the proposed method in Table 2, a direct comparison under the setup of Table 9 would be, in my opinion, the most critical.

---

> > > ### Author Response · Authors · 2025-11-18
> > > **Follow-up for Reviewer CRQ3 [W2]**
> > >
> > > We sincerely appreciate your fast response and the valuable feedback you have provided, which has enhanced our understanding of your suggestions and will be instrumental in improving our manuscript.
> > >
> > > There are two reasons why STIR was not included as a baseline in `Task 3`. First and foremost, as stated in `Lines 468–470`, we adhered to the experimental setup of SpikeGS, including the selection of baselines, datasets, and the training and testing pipeline. Since STIR was not included in the original SpikeGS framework, we maintained consistency with this established protocol. Second, baselines such as SpikeGS and DeblurGS natively support RGB multi-view inputs and produce deblurred RGB scene reconstructions. In contrast, STIR does not inherently support RGB input. To apply STIR in this context, one would need to independently process spike streams generated from the R, G, and B channels to predict dense, clear frames, then merge these into a single RGB image as input. We have done this procedure and the comparative analysis is provided accordingly.
> > >
> > > **Dataset: Blender Objects**
> > > | Methods | PSNR | SSIM |
> > > | :--- | :---: | :---: |
> > > | STIR+3DGS | 29.01 | 0.91 |
> > > | **SpikeGen+3DGS** | **31.81** | **0.94** |
> > >
> > > As demonstrated, the performance of STIR is somewhat affected by this inconsistency. However, from another perspective, it reflects that the decoupled dual-modality pre-training strategy in our SpikeGen provides greater flexibility for adaptation to diverse downstream tasks, aligning—albeit to a certain extent—with the human visual system.
> > >
> > > Once again, we sincerely appreciate your professional review. We hope that this follow-up has fully addressed any remaining concerns, and we look forward to your continued support.

---

> > > > ### Comment · Reviewer_CRQ3 · 2025-11-19
> > > >
> > > > Thank you for the additional clarification and experiment results. All my questions have been addressed. I've updated my grading. Please remember to revise the final manuscript to include these clarifications and addtional results.

---

> > > > > ### Author Response · Authors · 2025-11-20
> > > > > **Reply for Reviewer CRQ3**
> > > > >
> > > > > Of course! We are pleased to incorporate these updates into our revised PDF. With the valuable suggestions from you and the other reviewers, we believe our manuscript will be significantly improved.
> > > > >
> > > > > Last but not least, we sincerely appreciate your prompt response and support.

---

### Official Review · Reviewer_QTzd · 2025-10-29

**Soundness:** 3
**Presentation:** 3
**Contribution:** 2
**Rating:** 6
**Confidence:** 2

**Summary:**

This paper proposes a framework called SpikeGen, which takes a blurry image and the corresponding spike stream as input to generate a clear image. The authors evaluated the method on three downstream task datasets, demonstrating its effectiveness.

**Strengths:**

1. The manuscript is well-structured, the figures are clear, the layout is appropriate, and the language flows smoothly.

2. The approach is bio-inspired and well-justified.

3. The experimental evaluation is sufficient and the results demonstrated are promising.

**Weaknesses:**

1. The paper's citation format needs to be revised.

2. In Figure 2, I believe the blurry RGB latent and the Clear RGB latent should show differences (or be visually distinct), instead of being represented by the same shape and color.

**Questions:**

1. In lines 276 to 280, the authors describe the calculation process for the faded image $I_{faded}$. However, I could not seem to find where this result is applied within the method and Figure 2.

2. In line 312, the authors claim their model was pre-trained on ImageNet. However, that dataset only includes clear images. I am curious how the blurry images and the spike streams were obtained, especially the latter.

3. Based on the formula in line 270, the hyperparameter $\gamma$ takes a value of 0 or 1 with high probability (>60%). Does such a modality drop rate seem too high? This seems to indicate that the model mainly receives single-modality input during the training process.

4. How are the pixel-space similarity measures performed during fine-tuning with limited data?

---

> ### Author Response · Authors · 2025-11-17
> **Reply for Reviewer QTzd [W1,2, Q1,2,3]**
>
> We sincerely thank Reviewer QTzd for the positive feedback and for acknowledging the effort we invested in our paper's clarity, presentation, and motivation.
>
> Our work indeed spans multiple domains—from human visual systems and neuromorphic hardware to latent generative models. We spent a significant amount of time refining the manuscript, figures, and overall presentation to ensure our motivation was clear to readers from diverse backgrounds. We are delighted that this **effort was recognized, as all reviewers**, including you, found our paper well-structured and clear.
> We also appreciate the insightful questions, which highlight areas we can clarify. We will address each point below:
> ## Weaknesses 1 & 2 (Citation format and Figure clarity):
> Thank you for these suggestions. We will updated the manuscript to correct the citation formatting. We will also revised `Figure 2` to improve the visual distinction between different latent tokens, as you recommended.
> ## Question 1 (Color faded training strategy):
> This is an excellent question that points to a key part of our pre-training strategy. The $I_{faded}$ (described in `lines 272-280`) is used as the **ground-truth supervision signal** for the diffusion loss during pre-training.
>
> In `Figure 2`, for simplicity and clarity of the pipeline, the arrow from the "gamma generator" points directly to the "Clear RGB Latent". More precisely, the $\gamma$ value determines the degree of fading applied to the clear RGB image before it is encoded by the VAE.
> The process is:
> 1. Sample a $\gamma$ from the gamma generator.
> 2. Compute the faded image: $I_{faded} = (1-\gamma) \cdot I_{clear} + \gamma \cdot I_{gray}$.
> 3. Encode this faded image to get the ground-truth latent: $z_{faded} = VAE(I_{faded})$.
> 4. The diffusion loss is then computed between the model's prediction and this $z_{faded}$.
>
> This ensures that when the input modality is biased towards "spike only" ($\gamma \to 1$), the model is supervised by a (mostly) grayscale latent, encouraging it to focus on reconstructing texture and structure from the spike information, rather than hallucinating colors.
> ## Question 2 (Pre-train data generation):
> Due to the page limit, we placed the detailed methodology for synthesizing the blurry image and spike stream pairs in `Appendix Section A.2`, as referenced in `Section 4.1`.
>
> In short, we built upon the principles presented in the published SpikingSim simulator **[1]** and implemented a dataloader (available in our `supplementary material`) with definable parameters. For each clear RGB image from ImageNet:
> 1. We apply a motion kernel (e.g., $40 \times 40$) to generate the blurry image.
> 2. We compute a normalized photon sampling probability map based on the clear image's pixel intensities and then sample binary spike frames from this probability map over a simulated time window to generate the spike stream.
> ## Question 3 (Distribution of the $\gamma$ hyperparameter):
> Thank you for this very sharp observation regarding the $\gamma$ distribution. We acknowledge that our chosen truncated Gaussian $\mathcal{N}_{[0,1]}(\mu=0.5, \sigma^2=1)$ does sample near 0 or 1 by chance. However
> , this is, in fact, a **deliberate design** choice to achieve a balance between fusion specialization and general robustness.
> 1. **Primary Goal (Fusion)**: Our primary goal is to excel at dual-modality fusion. Our chosen distribution supports this, as the highest probability density (the mode) is still at the center, $\gamma=0.5$.
> 2. **Secondary Goal (Generality)**: In order to handle real-world scenarios where one modality might be weak or missing (e.g., our `Task 2`: Dense Frame Reconstruction). Our distribution ensures the model also encounters these single-modality-dominant scenarios ($\gamma \approx 0$ or $\gamma \approx 1$).
>
> This pursuit of generality is also why we designed the complementary **'color fade'** strategy (`Line 272-280`). This strategy explicitly adapts the learning objective based on $\gamma$: when the model is asked to rely on spikes ($\gamma \to 1$), the ground-truth is faded to grayscale, allowing the model to focus on texture reconstruction rather than color prediction.
>
> The success of this unified strategy is demonstrated throughout our experiments. As seen in `Figure 3` and `Figure 8`, our single pre-trained model **does not collapse when given only single-modality** inputs; it still produces reasonable results (e.g., 'SpikeGen (w/o RGB)'), even though the dual-modality performance is, as expected, optimal.
> If we had chosen a distribution with a very low dropout rate (e.g., a much smaller $\sigma$), the model would specialize only in fusion, and this crucial generality would be compromised.

---

> ### Author Response · Authors · 2025-11-17
> **Reply for Reviewer QTzd [Q4]**
>
> ## Question 4 (Fine-tune Strategy):
> As we mentioned in `Section 3.3`, relying only on latent-space supervision can lead to underperformance when downstream-task data is limited.
> Therefore, we wish to introduce pixel-space loss back during fine-tune. However, since in some case we lack clear RGB ground truth in real-world dataset (e.g, momVidar2021 **[2]** dataset in our `Task 2`), we cannot use standard losses like MSE or L1.
> Instead, we use Spike-Alignment Loss:
> 1. We take the model's predicted (decoded) RGB image, $I_{pred}$.
> 2. We pass $I_{pred}$ through our spike simulator to generate a synthetic spike stream, $P_{pred}$.
> 3. We then compute the loss between this predicted spike stream ($P_{pred}$) and the ground-truth spike stream (the one captured by the camera).
>
> This allows the spike stream itself to provide a form of weak supervision in the pixel space, guiding the model to produce images that are consistent with the observed motion and intensity changes, even without a clear RGB target.
>
> We hope these clarifications fully address your questions. We thank you again for your constructive review and positive evaluation.
> ## Reference
> [1] Zhao, Junwei, et al. "Spikingsim: A bio-inspired spiking simulator." 2022 IEEE International Symposium on Circuits and Systems (ISCAS). IEEE, 2022.
>
> [2] Zhu, Lin, et al. "Retina-like visual image reconstruction via spiking neural model." Proceedings of the IEEE/CVF Conference on Computer Vision and Pattern Recognition. 2020.

---

> > ### Comment · Reviewer_QTzd · 2025-11-26
> >
> > Thank you for your response and the detailed explanations. After reading your reply, I now have a clearer understanding of both the questions and the methods you proposed. In my view, the overall approach seems sound, and the evaluations you conducted seem thorough. I believe the paper indeed offers some new insights, but since I am not familiar with the related work, I am unable to assess its novelty.
> >
> > Based on this, I have slightly more confidence in the score I assigned (from 2 to 3).

---

> > > ### Author Response · Authors · 2025-11-26
> > > **Reply for Reviewer QTzd**
> > >
> > > Thank you for raising the confidence score! We are pleased to note that the unclear part in our initial submission has been resolved, which prove a productive discussion between you and us.
> > >
> > > Regarding the novelty of SpikeGen, we understand your cautious decision due to your unfamiliarity with the area. Nevertheless, we remain quite confident as other reviewers have commented our method as: "Elegant and efficient" (`Reviewer 146W`); "efficient, effective and original" (`Reviewer xKds`); "innovative and effective" (`Reviewer CRQ3`). These positive feedbacks may further support your judgment.

---

### Official Review · Reviewer_xKds · 2025-10-30

**Soundness:** 3
**Presentation:** 3
**Contribution:** 3
**Rating:** 4
**Confidence:** 4

**Summary:**

The paper proposes SpikeGen, a novel latent generative framework for decoupled visual representation learning between RGB and spike modalities. It performs diffusion modeling in the latent space, combining VAE-encoded representations with a per-token diffusion mechanism to balance efficiency and effectiveness. The framework incorporates a configurable dual-modality latent pre-training mechanism and a spatio-temporal separable spike encoder for efficiently extracting temporal–spatial features from spike streams. Experiments conducted on multiple benchmark datasets, including REDS, GOPRO, VidarReal, and Blender-NeRF, show that SpikeGen surpasses existing methods across multiple metrics, demonstrating superior generalization and robustness.

**Strengths:**

This paper introduces latent variables into such tasks for the first time, enabling the model to achieve higher-level feature modeling while maintaining both efficiency and effectiveness. The proposed method demonstrates strong originality, reliable and comprehensive experimental results, and clear exposition, making it an contribution to multimodal visual representation learning.

**Weaknesses:**

1. The training cost of SpikeGen’s latent diffusion process is relatively expensive and requires substantial pre-training resources, which limits its reproducibility and practical deployability.

2. During the pre-training phase on the ImageNet dataset, the spike frame configuration involves randomly sampling 8 frames from 64 generated spike frames for each image. This setup may prevent the model from fully leveraging information from the spike modality, causing it to rely primarily on RGB inputs and thus partially degrade into a single-modality model.

3. Certain details in the paper are described ambiguously. In Section 3.2 (Spatial–Temporal Separable Spike Latent), the authors mention that the model generates temporal attention weights through two consecutive 1×1×1 3D convolutional layers to model the temporal dimension explicitly. However, this process lacks formal equations or explicit computational explanations, which affects the interpretability and reproducibility of the method.

4. Although the model achieves impressive results on synthetic and benchmark datasets, its validation on real-world event-based data remains limited, making it insufficient to fully demonstrate the model’s generalization capability under complex real-world conditions.

**Questions:**

1. During the pre-training phase on ImageNet, the model randomly samples only 8 frames from 64 generated spike frames. How do the authors ensure that such sparse temporal sampling can still effectively capture spike information? Have experiments with different sampling numbers (e.g., 16 or 32 frames) been conducted to verify that the model indeed utilizes spike information rather than primarily relying on RGB features?

2. In Section 3.2, the authors mention that the model generates temporal attention weights through two consecutive 1×1×1 3D convolutional layers. Could the authors supplement this part by explaining how these weights are computed and applied, to clarify their role in the feature fusion process?

3. The model is pre-trained on ImageNet using 8 A800 GPUs. Does the model's outstanding performance stem from the retraining advantage gained through ample computational resources, or from model innovation?

---

> ### Author Response · Authors · 2025-11-19
> **Reply for Reviewer xKds [W1,2, Q1]**
>
> We sincerely thank the reviewer for the positive evaluation **(Soundness, Presentation, and Contribution: Good)** and for recognizing our work’s **"strong originality," "reliable experimental results," and "contribution to multimodal visual representation learning."** These comments are highly encouraging.
>
> We understand that your current rating (4) stems from specific concerns such as training costs and sampling strategies. We address these points below to alleviate your concerns.
> ## Weakness 1 (Training cost, reproducibility, and deployment):
> The reviewer noted that the pre-training cost of SpikeGen is "relatively" high and suggested that this may hinder reproducibility and deployment. However, the connection between pre-training cost and reproducibility is not entirely clear to us. To clarify, we provide the following explanation based on our understanding:
> 1. **Pre-training Cost:** As acknowledged in your `Strength 1`, the core innovation of SpikeGen lies in achieving a high compression ratio through Latent Operations. Consequently, compared to self-supervised methods operating in the pixel space, our approach is inherently **more efficient**. The remianing concern regarding computational cost arises from the diffusion process. We recognize that a standard conditional DiT **[1]** can be cumbersome and inefficient for our purposes, as it requires repeated application of the full ViT backbone with **all tokens across multiple diffusion steps**. To address this, we adopt a MAR-based architecture (illustrated in Figure 2), which utilizes the ViT auto-regressively as an encoder, with subsequent diffusion steps managed by a **lightweight MLP head**. This efficiency advantage is further underscored by the results in Table 7, which demonstrate that our model achieves state-of-the-art performance **even without autoregressive steps**—enabling simultaneous generation of all tokens. This capability further distinguishes our method from DiT-based approaches in both computational efficiency and performance.
> 2. **Reproducibility:** We have provided full data processing pipelines and training (including pre-train and fine-tune) codes in the `supplementary material`. We also commit to releasing the pre-trained weights, ensuring that the community can fine-tune and use SpikeGen without incurring the pre-training cost.
> 3. **Edge Device Deployment:** We believe that for an academic article, especially in the case of a conference like ICLR with a page limit, it is necessary to have **prominent conclusions to drive the community**. Apparently, our conclusion is that for the Spike-RGB dual modality, Latent Generation Model is an **framework with great potential**. It has structure similar to the human visual system and demonstrates good generalization ability to alleviate the "sharpness trap". Regarding efficiency, we have also taken it into consideration. As mentioned above, Latent Operation itself is an efficient approach. Secondly, instead of choosing DiT, we select MAR to further improving the efficiency. Finally, we have further verified that the autoregression of the original MAR can also be dispensed with, which once again enhances the processing efficiency. We hope that these explorations of ours can inspire subsequent explorations of edge device deployment.
>
> **Device: 1\*NVIDIA A100 80G | Resolution: 256\*256**
> | Backbone | time/256 images | Test PSNR (ImageNet) |
> | :--- | :---: | :---: |
> | DiT-B | 2min19s | 18.11 |
> | MAR-B | 1min32s | 18.79 |
> | **SpikeGen** | **2.7s** | **19.08** |
> ## Weakness 2 & Question 1 (temporal sampling 8/64):
> This is a great question. We address this from two perspectives:
> 1. **Self-Supervised Learning (SSL) Perspective**: Similar to MAE **[2]**, which uses a high masking ratio (e.g., 75%) to force the model to learn robust representations, we intentionally use sparse spikes (8 frames) and heavily blurred RGB. This "difficult" task forces the model to **integrate complementary information rather than relying on easy shortcuts**. To verify this, we conducted an ablation study where we increased the pre-training spike frames to 32. As shown below, increasing frame density not only increased computational cost but degraded transfer performance on deblurring task, suggesting overfitting.
> 2. **SpikeGen Pre-training Perspective:** You raised a valid concern that the model might ignore the spike modality. However, this is precisely what our Random Modality Dropout ($\gamma$) and Color Fade strategies (`Section 3.2`) prevent. During pre-training, we sample a $\gamma$ value to control the fusion weight. Accordingly, we fade the ground-truth RGB to grayscale. This preventing the model from collapsing into a pure-RGB modality. The successful single-modality results (SpikeGen w/o RGB) in `Figures 3, 4, 7, 9` and `Tables 2, 8` confirm that our **model does not collapse without RGB input**.
>
> **Dataset: GOPRO | $V_{th}$ = 1**
> |Frame #| PSNR |
> | :---: | :---: |
> | 32 | 26.99 |
> | **8** | **29.30** |

---

> ### Author Response · Authors · 2025-11-19
> **Reply for Reviewer xKds [W3,4, Q2,3]**
>
> ## Weakness 3 & Question 2 (Temporal fusion detail)
> As mentioned in the response to `Weakness 1`, the specific model structure and implementation code for this part is provided in our `Appendix Section B.1` and `supplementary material`. Here, we will further elaborate on the temporal fusion process of the S3 encoder:
> 1. After multi-scale downsampling (block merge style), the raw spike stream expand to raw spike latent
> > [B, 1, T, H, W] > [B, 512, T/8, H/8, W/8]
> 2. We use two consecutive 1x1x1 3D convolutions to calculate the weights of the spike latent, with LeakyReLU activation in between.
> > [B, 512, T/8, H/8, W/8] > [B, 512, T/8, H/8, W/8].
> 3. Perform a weighted sum over the time dimension to obtain the spike latent after temporal dimension aggregation
> > [B, 512, T/8, H/8, W/8] > [B, 512, H/8, W/8].
> ## Weakness 4 (Real-world Evaluation)
> Here, we need to point out two of your mistakes. First, our article is **spike-based** rather than **event-based**. The principles of these two types of cameras are quite different (spikes are based on integration, while events are based on differentiation). Therefore, we cannot directly process event data with SpikeGen. Second, we have included the results on **real spike stream data**. `Task2`:Dense Frame Reconstruction in `Section 4.2` is designed to demonstrate the performance in this regard. We show the performance on the **momVidar2021** dataset (`Figure 1` and `Table 2`), which is real data collected by a spike camera. Here, we add an additional experiment based on real spike streams, using the RS-3D dataset from the article SpikeNVS **[3]**. This article belongs to our `Task 3`: Novel-View Synthesis task. We hope this result can further validate our article.
>
> | Method | DeblurGS | SpikeNVS | SpikeGS | SpikeGen (3DGS) |
> | :--- | :--- | :--- | :--- | :--- |
> | BRISQUE ↓ | 57.1 | 52.4 | 50.7 | **51.0** |
> | NIQE ↓ | 4.88 | 4.52 | 4.06 | **3.71** |
> ## Question 3 (Computational Resources vs. Model Innovation)
> Your question is very profound. However, we must first point out that, in our opinion, **the success of the entire framework cannot be attributed to the role of a single factor**. Simply put, if there is only a large amount of pre-training without considering the model structure, both the training cost and the pre-training return will be less than satisfactory. If we only design the model structure for specific tasks and ignore how to scale up to a large amount of data, it is also very difficult for the model to carry out pre-training to obtain stronger generalization ability.
>
> Regarding SpikeGen, our primary drive force was base on the observation that the functions of the rods and cones cells in the retina and how both information integrated in cortex. This inspired us to consider the natural disadvantages of the corresponding single-modal imaging (spike - spatial sparsity, RGB - temporal sparsity) as a "degraded" form of visual representation. Then, we use the time-tested self-supervised learning method based on degradation - restoration to pre-train a model.
>
> Under this **general idea**, in order to further improve the **efficiency and effectiveness** of SpikeGen, we selected the Latent Generative Model to avoid operations in the pixel space. Further, in terms of efficiency, our optimization path is from DiT to MAR, and then to the non-autoregressive MAR. In terms of effectiveness, our optimization path involves introducing Random modality dropout and color fade during the pre-training stage, as well as spike alignment during the fine-tuning stage. This complete chain of thoughts is necessary to complete our framework, and we also believe that this large framework can inspire the derivation of subsequent work.
>
> We hope that our response has provided you with a clearer understanding of our work. Once again, we sincerely appreciate your positive feedback and look forward to your reevaluation.
> ## Reference
> [1] Peebles, William, and Saining Xie. "Scalable diffusion models with transformers." Proceedings of the IEEE/CVF international conference on computer vision. 2023.
>
> [2] He, Kaiming, et al. "Masked autoencoders are scalable vision learners." Proceedings of the IEEE/CVF conference on computer vision and pattern recognition. 2022.
>
> [3] Dai, Gaole, et al. "Spikenvs: Enhancing novel view synthesis from blurry images via spike camera." arXiv preprint arXiv:2404.06710 (2024).

---

### Official Review · Reviewer_146W · 2025-10-31

**Soundness:** 2
**Presentation:** 3
**Contribution:** 2
**Rating:** 4
**Confidence:** 5

**Summary:**

This paper presents SpikeGen, a biologically inspired framework that mimics the rods–cones decoupling mechanism in human vision.
The model fuses spike streams (representing temporal luminance information) and RGB images (representing spatial–chromatic information) in a shared latent diffusion space, thereby achieving joint visual representation learning.

**Strengths:**

1 The paper's multimodal fusion in the latent space rather than the pixel space is elegant and computationally efficient. The approach avoids the need for spatially precise alignment between asynchronous spikes and frames.

2 The paper includes quantitative and visual comparisons on multiple datasets and modalities, covering both synthetic and real-world-like settings.

**Weaknesses:**

(A) Task–Method Mismatch

The model’s design is static latent diffusion, but some tasks (e.g., motion deblurring) inherently require explicit temporal modeling (e.g., flow, exposure trajectory).
SpikeGen’s temporal encoder uses only 3D convolutions, which might not capture fine-grained dynamics.

In novel-view synthesis, the model lacks geometric consistency constraints (e.g., ray-based volumetric modeling).
The improved perceptual quality may not correspond to true 3D structure preservation.

(B) Evaluation and Fairness Issues

SpikeGen is compared against single-modality baselines (e.g., SpkDeblurNet, DeblurGS) while itself using RGB+Spike dual input — an unfair comparison unless baselines are also dual-modality.

The absence of clear ablation studies (e.g., removing diffusion, removing spike input, removing γ-fusion) makes it difficult to identify the true source of performance gain.

Training details for each task (e.g., hyperparameters, dataset scale, latent dimensionality) are insufficient for reproducibility.

(C) Data Authenticity and Realism

Most spike data are synthetic (via SpikingSim) and do not represent real sensor noise, asynchronous pixel behavior, or refractory effects.

For each task, the paper lacks quantitative evaluation on real spike-camera datasets.

(D) Overly Perfect Results / Missing Uncertainty

SpikeGen outperforms all prior works on all metrics across all tasks — an unlikely scenario that raises concerns about overfitting or inconsistent training conditions.

The diffusion framework, by nature, introduces randomness and perceptual diversity; yet, the results show unrealistically consistent sharpness and color balance without variance analysis.

(E) Efficiency and Practicality

Despite operating in latent space, diffusion models remain computationally expensive.
The paper does not report inference speed or energy consumption, which are crucial in neuromorphic vision research that emphasizes efficiency.

**Questions:**

1. How are the spike encoder’s temporal windows determined across different frame rates or datasets?
   Is the encoder adaptive to spike density variations?

2. For the deblurring task, does SpikeGen explicitly model the exposure time or motion trajectory, or rely solely on spike event accumulation?

3. How is the latent diffusion conditioned on spike features?
   Is it concatenation, cross-attention, or a learned fusion layer?

4. How does the model generalize to **real spike data** (e.g., Vidar)?
   Have the authors tested domain adaptation or noise robustness?

---

> ### Author Response · Authors · 2025-11-19
> **Reply for Reviewer 146W [WA,B, Q1,2]**
>
> We appreciate the reviewer’s recognition of our method as "**elegant and computationally efficient**" and the acknowledgement of the **sufficiency of our experiments**, a view shared by all other reviewers. However, we noticed some misunderstandings in your comments. We would like to address these point-by-point below, hoping to clarify these aspects and assist you in re-evaluating our paper.
> ## Weakness A & Question 2 (Task–Method Mismatch)
> **Clarification on Architecture:**
> We respectfully clarify that our method is **not a "static latent diffusion" approach** as characterized. Furthermore, the S3 Encoder in SpikeGen is **not merely a structure solely utilizing 3D convolutions**. As indicated by its name—"Spatial-temporal Separable Spike Encoder"—our architecture employs a decoupled design that performs inter-frame fusion via temporal attention mechanisms, followed by intra-frame fusion. To be specific:
> * We utilize distinct 3D and 2D convolution operations during different fusion stage.
> * The initial consecutive 3D convolutions operate in a block-merge manner. This is designed to downsample the spatiotemporal information of the Spike Stream gradually with expanding dimension correspondingly.
> * The following 1x1x1 3D convolutions within the temporal attention mechanism differs from the block-merge 3D convolutions.
>
> Therefore, we respectfully disagree that this process can be summarized simply as a "3D convolutional mapping on the Spike stream."
>
> **Explicit Temporal Modeling:**
> Fundamentally, deblurring involves predicting a deconvolution pattern to recover static structures from motion. While information like optical flow can serve as an effective prior, it is **not a strict necessity**. In fact, most baselines we compared against (e.g., S-SDM, DeblurNeRF) do not utilize optical flow.
>
> Moreover, the **Spike Stream** itself inherently carries the "explicit temporal modeling" you mentioned. Given our clear motivation—designing a framework for dual-modal Spike-RGB processing inspired by the biological **rods and cones** mechanism—incorporating optical flow as a third modality would diverge from our original motivation and arguably dilute the specific contribution we aim to present.
>
> **Geometric Consistency Constraints:**
> As stated at the beginning of `Task 3`: Novel-View Synthesis, SpikeGen use a **two-stage method** in this task. Consequently, the geometric consistency constraint for 3D reconstruction is **inherently provided by 3DGS** in the second stage. Our experiments essentially demonstrate that the deblurring in Stage 1 yields sufficiently high quality to allow a two-stage pipeline to rival one-stage methods. For a detailed discussion on the trade-offs between one-stage and two-stage approaches, please refer to our response to **Reviewer CRQ3 (Weakness 1)**.
> ## Weakness B & Question 1 (Evaluation Fairness)
> We demonstrate the validity and fairness of our experimental settings through the following three points:
> 1. **Adherence to Established Experimental Protocols:** Every task setting in our work closely aligns with experimental protocols from recently published literature to ensure fair comparison:
> * `Task 1`: Conditional Image/Video Deblurring. We followed the experimental setup of S-SDM (NeurIPS 2024), which uses SpkDeblurNet as a baseline. We respectfully wish to clarify a misunderstanding in your comment: SpkDeblurNet actually utilizes a dual-modal input (RGB + Spike Stream), **not a single modality.** The true single-modal baseline in this context is BiT (CVPR 2023), which takes **pure RGB** images as input. We have explicitly annotated whether each baseline is single-modal or multi-modal in `Table 1`.
> * `Task 2`: Dense Frame Reconstruction. We followed the setup of STIR (NeurIPS 2024). In this task, all methods utilize a **single Spike modality**, ensuring that the comparison is strictly fair.
> * `Task 3`: Novel-View Synthesis. We followed the setup of SpikeGS (AAAI 2025). Similar to Task 1, we included both dual-modal (SpikeGS) and single-modal (DeblurGS) methods, with clear indications provided in the paper.
> 2. **Comprehensive Ablation Studies:** We have conducted extensive ablation studies **covering** the aspects you mentioned:
> * Removing Spike/RGB Inputs (i.e., $\gamma$ ablation)
> > Please refer to `Figures 3, 7, 8` and `Table 8`.
> * Latent Diffusion Framework (MAR) Sampling Steps
> > Please refer to `Figure 6` and `Table 7`.
> * Additional Ablations
> > Please refer to `Appendix Sections C.1-5`
> 3. **Hyperparameters and Reproducibility:** Details regarding pre-training and fine-tuning hyperparameters are not only described within each paragraph of `Section 4` but are also comprehensively listed in `Appendix Section B` (covering the **dataset scale** and **latent dimensionality** you requested). Furthermore, we have provided our source code in the `supplementary material`. We trust that these resources are sufficient to resolve your concerns regarding implementation details and reproducibility.

---

> ### Author Response · Authors · 2025-11-19
> **Reply for Reviewer 146W [WC,D, Q4]**
>
> ## Weakness C & Question 4 (Data Authenticity and Realism)
> The choice of using synthetic data during pre-training is primarily limited by the current **absence of a large scale real-world Spike camera dataset** comparable to ImageNet. However, training on synthetic data is a widely accepted practice within this community. For instance, the published works we compare against, such as S-SDM and STIR, both utilize synthetic data for training and testing. We have simply scaled this approach up to an ImageNet-level volume, which indirectly demonstrates the **scalability** of our method.
>
> We fully acknowledge the inevitable domain gap between real and synthetic data. Consequently, we have ensured the inclusion of experiments based on real spike datasets. specifically regarding the quantitative experiments you mentioned:
> * `Figure 1` (Teaser Figure): Showcases performance on the **momVidar2021** dataset.
> * `Task 2` (Dense Frame Reconstruction): we use `Table 2` to provide further detailed comparisons with other baselines on real data.
>
> We have added an additional comparison on a real-world dataset for `Task 3: Novel-View Synthesis**, utilizing the RS-3D dataset from SpikeNVS **[1]**.
> | Method | DeblurGS | SpikeNVS | SpikeGS | SpikeGen (3DGS) |
> | :--- | :--- | :--- | :--- | :--- |
> | BRISQUE ↓ | 57.1 | 52.4 | 50.7 | **51.0** |
> | NIQE ↓ | 4.88 | 4.52 | 4.06 | **3.71** |
>
> ## Weakness D & Question 4 (Results Missing Uncertainty)
> First, we wish to clarify that we do not claim SpikeGen outperforms previous methods **across all tasks and all metrics**.
> Our most direct competitor is S-SDM (pixel-space pre-training framework), as detailed in our `Section 1` (Introduction).
>
> If you examine our experiments closely, you will observe that our optimal performance is in `Task 1`: Conditional Image/Video Deblurring. This is because this task perfectly aligns with our pre-training self-supervised objective (dual-modal fusion to restore clear images), leading to significant advantages across different settings.
>
> In `Task 2`:Dense Frame Reconstruction and `Task 3`:Novel-View Synthesis, SpikeGen does face certain limitations due to input deviation (Task 2 relies solely on the single Spike modality) and output deviation (Task 3 requires 3D scene reconstruction). To achieve performance **comparable** to domain-specific models, we employed additional adaptations (e.g., using TFP results as pseudo-grayscale images, employing 3DGS for two-stage reconstruction).
>
> The showcasing of these tasks was to demonstrate the **versatility** of our framework in handling different tasks via additional modules. We honestly reported the performance drop when these extensions are ablated in `Table 8` and `Figure 8`.
>
> Regarding your comment that our visualizations show "unrealistically consistent sharpness and color balance," could you kindly specify which Figure numbers you are referring to?
>
> We believe we have demonstrated **not only the optimal results** of SpikeGen under dual-modal conditions **but also the results under single-modal** input for the same images (see `Figures 3, 7`). We believe this intuitively showcases the **perceptual diversity** of our model. Furthermore, other reviewers have found our evaluation to be **reliable** (Reviewer xKds), **sufficient** (Reviewer QTzd), and **consistent** (Reviewer CRQ3).

---

> ### Author Response · Authors · 2025-11-19
> **Reply for Reviewer 146W [WE, Q3]**
>
> ## Weakness E (Efficiency and Practicality)
> We fully acknowledge that naively applying diffusion models can impact computational efficiency. To address this, we implemented **three specific efficiency improvements** in SpikeGen:
> 1.  **Latent Space Operations:** We adopted a Latent Generative Model to avoid computationally expensive operations in pixel space (we appreciate that you also recognized this in `Strength 1`).
> 2.  **Lightweight Diffusion Head:** Unlike standard diffusion models (e.g., DiT) that utilize the entire backbone for the diffusion process, we drew inspiration from MAR (Masked Auto-Regressive). We confined the diffusion process to the final lightweight MLP head, which significantly boosts efficiency.
> 3.  **Removal of Autoregression:** Furthermore, we identified that the autoregressive process in MAR is likely a necessary design for text/class-to-image generation. However, since ours is an image processing task, we experimentally removed this autoregressive step. As shown in `Figure 6` and `Table 7`, this modification not only maintained performance but further improved processing efficiency.
>
> Every field faces a **trade-off between efficiency and effectiveness**. While our work leans towards utilizing modern frameworks to improve **effectiveness**, we have strived to optimize efficiency through architectural choices and pipeline refinements. By way of analogy, other efficiency-sensitive fields, such as object detection, have successfully transitioned from CNN-dominated frameworks (e.g., R-CNN **[2]** series) to Transformer-dominated frameworks (e.g., DETR **[3]** series), a shift widely accepted by the community.
>
> We have provided an ablation study demonstrating the specific efficiency gains from these three improvements for your reference (please also refer to our response to **Reviewer CRQ3, Weakness 3**).
>
> **Device: 1\*NVIDIA A100 80G | Resolution: 256\*256**
> | Backbone | time/256 images | Cost List | Test PSNR (ImageNet) |
> | :--- | :---: | :--- | :---: |
> | DiT-B | 2min19s | (100\*ViT Forward)\*256 tokens | 18.11 |
> | MAR-B | 1min32s | (64\*ViT+100\*MLP Forward)\*gradually to 256 tokens | 18.79 |
> | **SpikeGen** | **2.7s** | **(1\*ViT+100\*MLP Forward)\*256 tokens** | **19.08** |
>
> ## Question 3 (How to condition on spike features)
> As mentioned above, our diffusion process is located within the lightweight MLP head. This MLP receives the Spike-RGB features—which have already been fully fused by the ViT backbone—as conditions. Information is injected using **Adaptive Layer Norm (adaLN)**, following the standard MAR framework.
> Prior to entering the ViT backbone:
> 1.  The **S3 Encoder** and **VAE** encode the Spike and RGB information, respectively.
> 2.  These are then fused via weighted summation base on the $\gamma$ value.
>
> This weighted input mechanism allows us to easily adjust the model's bias by controlling $\gamma$ after pre-training. This flexibility not only adapts the model to dual-modal downstream tasks (`Task 1,3`) but also effectively prevents model collapse when handling single-modal inputs (`Task 2`).
>
> ## Reference
> [1] Dai, Gaole, et al. "Spikenvs: Enhancing novel view synthesis from blurry images via spike camera." arXiv preprint arXiv:2404.06710 (2024).
>
> [2] Girshick, Ross. "Fast r-cnn." Proceedings of the IEEE international conference on computer vision. 2015.
>
> [3] Zhu, Xizhou, et al. "Deformable detr: Deformable transformers for end-to-end object detection." arXiv preprint arXiv:2010.04159 (2020).

---

### Author Response · Authors · 2025-11-20
**Global Response [Update of PDF & Point-by-Point Reply]**

First and foremost, we would like to express our sincere gratitude to the Reviewers, Area Chairs, and Program Chairs for their time and dedicated effort. We have carefully studied **all comments** and have provided detailed **point-by-point** responses under each Reviewers.

We are encouraged not only by the positive scores but, more importantly, by the recognition of our work from the reviewers across three key dimensions:
* **Method:** "Elegant and efficient" (`Reviewer 146W`); "efficient, effective and original" (`Reviewer xKds`); "well-justified" (`Reviewer QTzd`); "innovative and effective" (`Reviewer CRQ3`).
* **Evaluation:** "Multiple datasets and modalities" (`Reviewer 146W`); "reliable and comprehensive" (`Reviewer xKds`); "sufficient and promising" (`Reviewer QTzd`); "consistent" (`Reviewer CRQ3`).
* **Presentation:** "Clear" (`Reviewer xKds`); "well-structured" (`Reviewer QTzd`); "well-organized and informative" (`Reviewer CRQ3`).

We also acknowledge the limitations in our initial submission. For instance, due to space constraints, we were unable to fully elaborate on our **efficiency optimizations**. This led to a misconception among some reviewers that our method incurs high computational costs simply because it is based on latent diffusion. In reality, while prioritizing performance ("effectiveness"), we have also improved "efficiency" through rigorous architectural choices and designs. We have detailed these optimizations in our responses and have prepared a **revised PDF** (with updates highlighted in **orange**) for your reference.

We are grateful for the active engagement of reviewers such as **Reviewer CRQ3**, who has already entered into multi-round discussions with us and has further acknowledged our contributions. We sincerely hope to engage in similarly constructive dialogue with other reviewers, as we believe this exchange is vital for improving the quality of our work.

Best regards,

**Authors of Submission 6977**

---

### Comment · Area_Chair_Jrco · 2025-11-26

Dear reviewers,

Please check the author's reply. Feel free to raise any questions or start a discussion, regardless of whether you will change the score.

Your AC.

---

### Author Response · Authors · 2025-11-29
**Summary from the Author**

# Dear Area Chair,

We deeply regret the unprecedented information leakage incident that has affected ICLR this year. We fully understand that the sudden surge in workload and the resulting disruption must be incredibly challenging for you and the other organizers.

As authors, we wish to provide a concise summary of the **"Initial Review,"** **"Our Rebuttal,"** and the **"Discussion Timeline"** to assist in your decision-making process.

## **Initial Review**
A brief summary of our SpikeGen
> This paper presents SpikeGen, a biologically inspired framework that mimics the rods–cones decoupling mechanism in human vision. The model fuses spike streams (representing temporal luminance information) and RGB images (representing spatial–chromatic information) in a shared latent diffusion space, thereby achieving joint visual representation learning (`146W, CRQ3`). It performs diffusion modeling in the latent space, combining VAE-encoded representations with a per-token diffusion mechanism to balance efficiency and effectiveness (`xKds, QTzd`). The framework incorporates a configurable dual-modality latent pre-training mechanism and a spatio-temporal separable spike encoder for efficiently extracting temporal–spatial features from spike streams. Experiments conducted on multiple benchmark datasets, including REDS, GOPRO, VidarReal, and Blender-NeRF, show that SpikeGen surpasses existing methods across multiple metrics, demonstrating superior generalization and robustness (`146W, CRQ3, xKds, QTzd`).

We are encouraged not only by the **positive scores** (`QTzd, CRQ3`) but, more importantly, by the **consistent recognition** of our work across three key dimensions:

* **Method:** Described as "Elegant and computationally efficient" (`146W`), "efficient, effective and original" (`xKds`), "well-justified" (`QTzd`), and "innovative and effective" (`CRQ3`).
* **Evaluation:** Acknowledged for covering "Multiple datasets and modalities" (`146W`) and being "reliable and comprehensive" (`xKds`), "sufficient and promising" (`QTzd`), and "consistent" (`CRQ3`).
* **Presentation:** Praised as "Clear" (`xKds`), "well-structured" (`QTzd`), and "well-organized and informative" (`CRQ3`).

## **Our Rebuttal**
We have provided a **point-by-point response** for **each reviewer** and prepared a **revised PDF** (with updates highlighted in orange) for reference. Please feel free to review them.

In our rebuttal, we endeavor to offer more **detailed explanations** regarding specific topics that might not have been thoroughly discussed in our initial submission, primarily due to the page limit. For instance, our efficiency optimizations fall into this category. Here, we would like to clarify that although we prioritize performance ("effectiveness") by scaling up pre-training and adopting modern latent generative backbone, SpikeGen is not "bulky" and has attained high efficiency through meticulous architectural selections (avoiding diffusion on ViT) and modifications (removing auto-regression). Additionally, we have evaluated our framework using real spike camera datasets, such as momVidar2021 and RS-3D, to showcase SpikeGen's performance during deployment. Please check the listed experiments in our response for more.

## **Discussion Timeline**

Notably, prior to the major leakage event (**Nov 27, 2025**), we had already engaged in active discussions with two reviewers and received further recognition from them.

| Reviewer ID | `CRQ3` | `QTzd` | `xKds` | `146W` |
| :--- | :--- | :--- | :---: | :---: |
| **Initial Score** | 6 | 6 | 4 | 4 |
| **If Discuss** | ✅ ([see revision history](https://openreview.net/revisions?id=DEfNwujoZ5)) | ✅ ([see revision history](https://openreview.net/revisions?id=3NrvH55NmA)) | ❌ | ❌ |
| **Reaction** | **Score:** 6 ⬆️ `8`  | **Confidence:** 2 ⬆️ `3` | - | - |
| **Date (Last Comment)** | Nov 20, 2025 | Nov 26, 2025 | - | - |

We also present the final comment of the reviewers here:
1. `CRQ3`
> Thank you for the additional clarification and experiment results. **All my questions have been addressed. I've updated my grading**.
2. `QTzd`
> Thank you for your response and the detailed explanations. After reading your reply, I now have a clearer understanding of both the questions and the methods you proposed. In my view, the **overall approach seems sound, and the evaluations you conducted seem thorough**. I believe the paper indeed offers some **new insights**.

Unfortunately, due to the incident, we were unable to continue the discussion phase with `xKds` and `146W`, preventing them from responding to our rebuttal. However, we are confident that the quality of our manuscript, combined with our detailed point-by-point responses, **is sufficient to resolve the queries raised by these reviewers.**

We trust in your professionalism and your ability to make an objective decision despite these emergency circumstances. Thank you for your dedication and hard work.

Sincerely,

**Authors of Submission 6977**

---

### Meta-Review · Area_Chair_TpnF · 2026-01-05

**Summary:**

The paper combines RGB and spike camera inputs into a common latent space and train a diffusion model to tackle tasks like deblurring.

The paper received initial mixed reviews. Reviewers praised the originality, efficiency, quality of exposition, the experiments and results. Concerns were raised about the usefulness of the extra modality, lacking ablations, and lacking experiments on real-world data.

The rebuttal clarified several points. Some of the concerns raised were already resolved in the submission. While some minor concerns still remain, I believe it passes the bar for ICLR, thus I recommend acceptance.

**Reviewer Concerns:**

The most serious concerns were about 1) how much does the extra modality really helps (146W, QTzd) and 2) lack of real-world experiments (146W, xKd).

1) The rebuttal listed a number of figures and tables in the paper showing the effects of the extra modality but many of those are only qualitative. For example, Fig 3 shows deblurring results for SpikeGen w/o Spike but I could not find corresponding quantitative results to show how much the adding the spike modality improves over that.

2) The lack of real-world experiments was rebutted by pointing out results on momVidar2021 in table 2 and adding results on RS-3D. The reviewers concerns was that most of the experiments use synthetic spike data (which is also used for training), and only one uses real data, in which the submission outperforms the best baseline in one metric but not in the other. Adding a single extra experiment with real data partially address the concerns; some reviewers might have been expecting more.

Other minor concerns were resolved, either by pointing to results already in the paper or providing extra results in the rebuttal.

**Reviewer Scores:**

146W's concerns were partially addressed as I explained in point 1) before. I believe they would maintain or slightly increase their score.

xKds's main concerns were mostly addressed (see my point 2) above). I believe they would increase their score, but there is a chance that they would be expecting more real-world data and thus would maintain the score.

QTzd and CRQ3 have initial favorable recommendations, and the rebuttal clarified many of their points so they would either increase slightly or maintain the scores.

---

### Decision · Program_Chairs · 2026-01-26

Accept (Poster)